# Impact of lifetime opioid exposure on arterial stiffness and vascular age: cross-sectional and longitudinal studies in men and women

Albert Stuart Reece, Gary Kenneth Hulse

▶ Prepublication history and additional material is available. To view please visit the journal (http://dx.doi.org/10.1136/bmjopen-2013-004521).

School of Psychiatry and Clinical Neurosciences, University of Western Australia, Crawley, Western Australia, Australia

**Correspondence to**
Dr Albert Stuart Reece;
sreece@bigpond.net.au

## ABSTRACT

**Objective:** To characterise and compare the potentiation of arterial stiffness and vascular ageing by opioids in men and women.

**Design:** Cross-sectional and longitudinal studies of 576 clinical controls and 687 opioid-dependent patients (ODP) on 710 and 1305 occasions, respectively, over a total of 2382 days (6.52 years), 2006–2011. Methodology Radial pulse wave analysis with Atcor SphygmoCor system (Sydney).

**Setting:** Primary care.

**Participants:** *Controls*: General practice patients with non-cardiovascular disorders, and university student controls. *ODP*: Patients undergoing clinical management of their opioid dependence. Controls had lower chronological ages (CAs) than ODP (30.0±0.5 vs 34.5±0.3, mean±SEM, p<0.0001). 69.6% and 67.7% participants were men, and 16% and 92.3% were smokers (p<0.0001) for controls and ODP, respectively. 86.3%, 10.3% and 3.4% of ODP were treated with buprenorphine (6.98±0.21 mg), methadone (63.04±4.01 mg) or implant naltrexone, respectively. Body mass index (BMI) was depressed in ODP.

**Interventions:** Nil.

**Primary outcome measures:** Vascular Reference Age (RA) and the ratio of vascular age to chronological age (RA/CA).

**Secondary outcome measures:** Arterial stiffness including Augmentation Index.

**Results:** After BMI adjustment, RA in ODP was higher as a function of CA and of time (both p<0.05). Modelled mean RA in control and ODP was 35.6 and 36.3 years (+1.97%) in men, and 34.5 and 39.2 years (+13.43%) in women, respectively. Changes in RA and major arterial stiffness indices were worse in women both as a factor (p = 0.0036) and in interaction with CA (p = 0.0040). Quadratic, cubic and quartic functions of opioid exposure duration outperformed linear models with RA/CA over CA and over time. The opioid dose–response relationship persisted longitudinally after multiple adjustments from p=0.0013 in men and p=0.0073 in women.

**Conclusions:** Data show that lifetime opioid exposure, an interactive cardiovascular risk factor, particularly in women, is related to linear, quadratic, cubic and quartic functions of treatment duration and is consistent with other literature of accelerated ageing in patients with OD.

### Strengths and limitations of this study

- With 576 controls and 687 OPD the large size of this study was a significant advantage, as was its combined cross-sectional and longitudinal design, its ability to control for many cardiovascular risk factors, the sex-specific analysis and the diverse output of cardiovascular parameters of the Atcor/SphygmoCor Pulse Wave Analysis technology employed.
- The use of detailed statistical modelling of relationships between factors, dose–response modelling, polynomial modelling in opioid exposure duration, inter-quartile trend effects and adjustment for known cardiovascular risk factors added further analytical power to the study.
- As the major study limitation was its lack of other measures of organism ageing and its lack of mechanistic and ageing medicine-specific biomarkers; further studies incorporating these refinements are indicated.

## INTRODUCTION

Opioid dependence is a condition of increasing public health importance.

Opioid dependence is associated with an impressive array of traditional and novel cardiovascular risk factors and associations including hypertension,[1] hyperlipidaemia,[2][3] tobacco consumption, increased body weight,[4] hyperglycaemia,[3] hyperfibrinogenaemia,[3] diffuse polyclonal gammopathy and immune stimulation,[5][6] elevated high sensitivity C reactive protein (CRP),[7] elevated cytokines,[8] relative hypercalcaemia and hyperphosphataemia[9] and reduced circulating endothelial progenitor cells.[10] Furthermore, opioid dependence has been associated with atherosclerotic disease in the coronary[11–18] and cerebral[14][18] circulations. In a 33 year review of an opioid maintenance programme in California in 2004, death from cardiovascular disease was noted to account for three times

as many years of life lost in the treatment group than in the general population.[18] In 2007 an Iranian surgical group identified that opium-dependent patients were at increased risk of requiring open coronary revascularisation with an OR of 1.8 after adjustment for other risk factors.[15] Moreover, in this series there was a dose–response relationship between the amount of opioids consumed and the severity of the coronary disease. It was also shown that coronary disease occurred 2 years earlier in opioid-dependent persons than in controls.[16] This group further identified that opium use was the strongest cardiovascular risk factor in their population of men, with effects greater than traditional risk factors.[17] The sex-specific finding in this study was felt to be related only to a reduced prevalence of opium consumption among women.

In reviewing over 800 fatal cases of opioid toxicity in Australia in 2006, Darke et al[12] found that 17% of decedents over the age of 44 years had coronary stenoses in excess of 75%. Similarly, when comparing 1193 postmortem cases of heroin and methadone toxicity, the Sydney group found 2–3 times elevated adjusted ORs of ventricular hypertrophy, and myocardial, interstitial and perivascular fibrosis in the methadone-treated group, consistent with the effects of more consistent opioid agonism in such patients.[11] In a 21-year follow-up of 42 676 patients undergoing maintenance opioid agonist treatment in Australia, including 425 998 person-years, Degenhardt et al[13] identified cardiovascular death as occurring 2.2 times as often as in the wider community. Indeed, in this study, elevated rates of death with severe organ dysfunction were noted which appeared to act as a backdrop to the very high rates of death by overdose generally reported in such cohorts (see Web appendix 6 of this citation).

In 2012, a prospective population-based study of 50 045 people, including 234 098 person-years, reported HRs of death from coronary artery disease (CAD) of 1.9-fold (95% CI 1.57 to 2.29) and from cerebrovascular disease (CeVD) of 1.68-fold (CI 1.29 to 2.18).[14] Both were worse in women (CAD: male adjusted HR (AHR) 1.58, female AHR 2.90; CeVD male AHR 1.49, female AHR 1.97). An earlier study from Iran did not confirm the cardiovascular salience of opioid exposure among women as compared to men, but the number of exposed women in this study was small, which likely compromised the power of the analysis to detect an effect.[17] Furthermore, recent reports of 1263 patients found elevated rates of vascular age in patients treated with methadone compared with other agents[19] and improved vascular ages accompanying the long-term drug-free condition.[20]

One large case control study found a dose–response relationship between opium exposure and the severity of angiographically assessed CAD in a diabetic population.[21]

Moreover, opioid patients dependent on short-acting opioids such as morphine, oxycodone and heroin

undergo the clinical syndrome of opioid withdrawal several times daily. This syndrome is a hyper-adrenergic state characterised by neurological and cardiovascular stimulation which in its more severe manifestations includes hypertension and tachycardia[22] and a reduction of subendocardial perfusion.[23] This negative aversive experience is one of the key reasons many opioid-dependent patients (ODP) continue to use narcotics.

Important as the above collected findings are, other evidence suggests that the impact of elevated vascular age has implications on the health and ageing of an organism as a whole beyond those limited primarily to the cardiovascular system. It has been repeatedly proposed that vascular ageing is a principal component of biological organismal ageing by virtue of the contribution made by cardiovascular causes to the overall death rate in developed nations, and also the importance of the vascular niche common to many stem cell beds.[24 25] This view closely accords with the findings in many of the above studies and particularly that by Degenhardt et al[13] that elevated rates of overdose occur on the background of advanced disease in other organs.[1 12 18] ODP have many stigmata of accelerated ageing including increased rates of hair greying, erosion of alveolar bone from chronic periodontitis,[26] osteoporosis,[27] chronic respiratory disease,[11 13 18] chronic renal disease,[11] diffuse senescent-mimetic immune stimulation,[5 6 28] diabetes,[3] neuropsychiatric disorders, multisystem disease,[11 12] reduced circulating stem cell counts[10] and many malignancies.[13 29] Thus the call has been made for geriatricians to be appointed to the care of ODP beyond the age of 50 years, due to the high prevalence of multisystem disease.[1]

The possibility therefore arises that an improved understanding of the increased and accelerated profile of degenerative disease seen in ODP may enhance our understanding, and in time, our management of age-related disorders in the rest of the community.

As this clinic sees ODP and the general community, we were ideally situated to compare these two groups. The availability of the mobile SphygmoCor system from Atcor in Sydney means that a host of central cardiovascular data is available to the clinician in a non-invasive and rapid manner, providing quantitation of important subclinical endophenotypes of central vascular stiffness and cardiovascular and therefore, organismal ageing. The central investigative question to be prospectively assessed was whether there were changes in major central cardiovascular parameters in the opioid-dependent group, and if so are they robust to adjustment for other confounding factors. Several earlier papers from our group have considered this issue in cross-sectional[30 31] and longitudinal[32 33] studies in each sex separately and drawn conclusions generally confirmatory of this hypothesis. It has been shown that the type of opioid pharmacotherapy affects the severity of opioid-related potentiation of arterial stiffness,[19] that induction of the drug-free state after implantable opioid antagonists reverses this finding[20]

and that opioid withdrawal is also associated with elevation of cardiovascular age and arterial stiffness parameters.[23] However a formal comparison of the effects by gender has not been possible in these earlier reports as they primarily considered single sex data. Gender comparison therefore forms the major focus of the present report which is a compilation of much of this earlier data.

## METHODS
### Patient selection
Control subjects (N=576) were recruited opportunistically from patients presenting for health examination checks for insurance or employment medical examinations or from patients presenting with minor health problems such as ear blockages, minor psychological disorders or university students. Patients with known cardiovascular disorders, such as hypertension, renal disease or diabetes or diseases known to perturb the CVS subacutely, such as infections or pregnancy, were excluded. ODP (N=687) were also recruited opportunistically and were under treatment with either methadone (N=71), buprenorphine (N=593) maintenance or naltrexone implant (N=23). All were sampled opportunistically at the time of their clinic visits between May 2006 and December 2012.

### Patient treatment
ODP maintained on methadone or buprenorphine were managed by accepted clinical algorithms by their usual health providers for their drug dependency. The naltrexone implants used for management of heroin dependence were manufactured by 'Go Medical' industries in Perth, Western Australia, under Commonwealth of Australia Therapeutic Goods Administration (TGA) Good Manufacturing Practice (GMP) conditions and administered under the compassionate access arrangements of the Special Access Scheme authorised by the TGA as described elsewhere.[34] Patients were recalled for retesting at 2 and 5 years postbaseline assessment, and assessed opportunistically. Any patient with a known chronic or subacute cardiovascular condition was excluded from participation. Frequency and quantity of substance use by substance type (inclusive heroin) was assessed at the initial time and subsequently. Other opioids were converted into morphine equivalents, and then into heroin equivalents at the rate of 1 g of street heroin=0.5 g morphine. The duration of opioid use was counted from the time of first use. If it became known that alcohol had been consumed in the few hours prior to the test, or that stimulants had been used in the 3 days prior or that patients were acutely affected by short-acting opioids, that day's studies were excluded.

### Pulse Wave Analysis studies
Patients were positioned supine for the performance of the test and allowed to rest for 5 min prior to the test.

Consumption of food, drink, alcohol or tobacco was not restricted, but patients were not allowed either to sleep or talk during the testing procedure. Pulse wave analysis (PWA) by applanation tonometry was performed on the right radial artery unless it was unavailable (due to trauma, surgery or congenital causes). The brachial blood pressure was taken in the opposite arm using the oscillometric Omron HEM 907 device. The data were collected and analysed by the SphygmoCor software. Patients were studied in quintuplicate, and adequate studies were averaged for that day. Acceptable studies had an operator index of greater than 70%, and were not inconclusive. Details of patients' prior drug use was also taken at the time of the study performance and entered into the database. Time in days was measured from the time of the first PWA study.

The SphygmoCor software used for the PWA studies calculates a number of major indices including Vascular Reference Age (RA), Chronological Age (CA), Central Augmentation Pressure at Heart Rate 75 (C_AP_HR7), the Central Augmentation Pressure/Pulse Height ratio at Heart Rate 75 (C_AGPH_HR75) also known as the Augmentation Index, Central Pulse Height (C_PH), Central Augmentation Load (C_AL), Central Augmentation Time Index (C_ATI), Peripheral-Central Pulse Pressure Amplification Ratio (PPAmpRatio), Maximal Peripheral Rise of dP/dT (P_MAX_DPDT), Central Systolic Pressure (C_SP), Central Diastolic Pressure (C_DP), Central End Systolic Pressure (C_ESP), Central Mean Pressure (C_MEANP), the Central Diastolic Time Index (C_DTI), the Central Tension Time Index (C_TTI) and an index of subendocardial perfusion known variously as the Subendocardial Perfusion Ratio (SEVR), the Central Stroke Volume Index (C_SVI) or the Buckberg ratio defined as C_TTI/ C_DTI.

### Statistics
Data are presented as the mean±SEM. Categorical data were studied using 'EpiInfo' 7.0.8.3 using the $\chi^2$ test obtained from the CDC, Atlanta, Georgia, or the Fisher's exact test if the numbers in a cell were less than 20. Continuous categorised data were analysed in 'Statistica' 7.1 from Statsoft, Oklahoma, using Student's t tests. T tests with separate variances were used where the Levene's test was significant. Continuous variables such as RA, CA, RA/CA, variously corrected RA, SP, weight, high-density lipoprotein (HDL), body mass index (BMI) and systolic pressure were log transformed in accordance with normality assumptions based on the Shapiro-Wilks test. High-sensitivity CRP and the heroin dose were arcsinh transformed which is a similar transform to logarithmic, but accepts arguments of zero. Time (as days) was measured from the time of the initial PWA study. Time since last cigarette was recorded and analysed as a factor with cut-off points at 30, 60, 120, 180 and >180 min and a non-smoking category. For some analyses, lifetime heroin exposure was cut into quartile

categories with divisions at 0, 0.01, 2.49, 6.59 and 350 g-years, respectively, yielding 583, 207, 226 and 227 patients in each group, respectively. Multiple regression was performed in 'R' 2.15.2 obtained from the University of Melbourne Central 'R' Archive Network mirror. Mean model estimates for various groups were prepared by inserting the mean parameter values into the untransformed final models. Multiple linear regression of truncated variables such as RA (minimum value of 20, maximum value of >80) was analysed using Tobit multiple regression techniques from the Applied Econometrics library in 'R' (AER). RAs of individual studies returned as >80 were assigned as 88. Time-dependent repeated measures analysis was performed using non-linear mixed effects (nlme) regression module in 'R' using restricted equation maximum likelihood techniques. Random effects were assigned to unity and the patient identification code. When nlme models were compared they were converted to maximum likelihood methods to allow ANOVA testing. In each case, model reduction was performed manually by the classical method involving removing the least significant term. The log-likelihood value was abbreviated to LogLik. Graphs were prepared with reshape2 and ggplot2. Where LOESS curves were fitted, the span used was 0.95. T tests were two-sided. $p < 0.05$ was considered significant.

## Ethical approval

All patients provided their informed consent to participate in the study and for their medical treatment. The study was approved by the Human Research Committee of the Southcity Family Medical Centre, Brisbane, Queensland, Australia, which is registered with the National Health and Medical Research Council.

## RESULTS

A total of 576 control patients were compared with 687 ODP. The mean ages were 30.0±0.5 and 34.5±0.3 (mean ±SEM) years (t=−8.262, df=1082.22, p<0.0001, see online supplementary table S1), with 401 (69.62%) and 465 (67.69%) men, respectively ($\chi^2$=0.54, p=0.46). In total, 15.97% and 92.29% of the control and ODP men smoked tobacco, respectively. ($\chi^2$=746.56, p<0.0001). The mean CA of the whole group was 32.5±0.3 years. The mean CA of men was 32.1±0.3 and of women 33.2 ±0.5 years. In total, 7.47% and 5.97% of the control and opioid-dependent groups were aged over 50 years, respectively ($\chi^2$=1.13, p=0.29). Among the men these ratios were 5.74% and 6.24% ($\chi^2$=0.1), and among the women they were 11.43% and 5.41% for the control and opioid-dependent groups, respectively ($\chi^2$=4.79, p=0.029, Fisher's exact test 0.19–0.98; pooled data, Corr. $\chi^2$=3.8, p=0.051). Racial stratification was not performed as around 90% of our population is of Caucasian background. Other baseline biometric, drug use and laboratory values are given in online supplementary table S1.

Patients in the control group and ODP were studied longitudinally on 710 and 1305 occasions, respectively. The period of repeat study, measured from the time of the first PWA study, was 624.11±18.74 days (range 1–2382 days; 495 (60.36%) studies performed beyond 360 days). Details of the numbers of repeat studies are given in online supplementary table S3.

Comparative PWA values are given in online supplementary table S3. Most augmentation values and the C_SP and C_ESP are noted to be higher in the opioid-dependent group consistent with their being an average 4.59 years older.

Figure 1 shows different vascular age and augmentation parameters corrected for various features against CA. This figure shows that BMI was lower in the opioid-dependent group than in the controls as a factor (factor est.=0.0719, est. SE=0.0327, t=2.197, p=0.0282) and in interaction with CA (BMI: CA est.=−0.0023, est. SE=0.0009, t=−2.350, p=0.0189; model F=34.94, df=3,1258, p<0.0001). In the longitudinal series BMI was lower as a factor (est.=−0.0174, df=812, p=0.0470; model Akaike Information Criteria (AIC)=−3722.55, LogLik= 1867.27). This result may be the effect of cumulative diversion of resources over time. Figure 2 shows similar indices over time. Online supplementary table S3 presents the results of the cross-sectional data where BMI corrected RA is regressed against CA and addictive status. In the whole group opioid-dependency status (p=0.0265) and the CA:addiction interaction (p= 0.0205) are significant. Similar results were found in each sex. Similar results were also found in the longitudinal study when the RA/CA ratio was regressed against CA, time and dependency status for opioid dependency as a factor (p=0.0102) and in interaction with CA (p=0.0028; details in online supplementary table S4).

Inclusion of tobacco exposure along with opioid exposure in these models did not alter these findings (data not shown).

As shown in supplementary figures S1 and S2, these changes were found to be more pronounced with CA in females (sex as a factor p = 0.0036; sex: CA interaction p=0.0040, supplementary table 5). These data are presented corrected for the changes in BMI in figures 3 and 4. Details for the significant differences in the cross-sectional dataset are presented in table 1. For the longitudinal dataset the data are given in supplementary table S6.

Figure 5 shows these data by sex with LOESS curves fitted (span=0.95). This strongly suggests that an inflection point is reached at about the age of 30 in women, which is not seen in men until about they are 40 years of age. This was quantified by dichotomising CA from age 41 in men and 29 in women. As shown in table 2, in men, in the older age category, the addictive status and their interaction was significant (p=0.0010, 0.0021 and 0.0008 respectively), while in women the older age: addiction interaction was significant (p=0.0195, analysis restricted to women <55 years).

**Figure 1**  Ageing and stiffness indices by chronological age by opioid dependency status. RA, Vascular Reference Age; CA, Chronological Age; C_AP_HR75, Central Augmentation Pressure at Heart Rate 75 bpm; C_AGPH_HR75, Central Augmentation Pressure/Pulse Height Ratio at Heart Rate 75 bpm.

The regression changes are quantified for some of the major indices of arterial stiffness in cross-sectional data in online supplementary table S6 (all p<0.05). Similar significantly different sex-specific data were found in the longitudinal dataset when the RA was regressed against CA, time, sex and addictive status (most p<0.02, see online supplementary table S7).

When the mean sex-specific values for CA and BMI are inserted into the equation of table 3, the mean vascular ages of the control groups among all patients, men and women were 35.42, 35.61 and 34.45 years, while the corresponding ages of the ODPs were 37.24, 36.31 and 39.18 years, representing incremental gains of 5.14%, 1.97% and 13.43%, respectively.

When quartiles are defined for lifetime heroin exposure the results shown in figure 6 are found. Similar effects longitudinally are shown in online supplementary figure 3. These results are quantitated in online

supplementary table S8 for cross-sectional and longitudinal data compared to non-exposed controls. In the cross-sectional analysis the RA/CA ratio was regressed against the CA and Opioid exposure quartile. In the longitudinal analysis the RA/CA ratio was regressed against time, CA and the opioid exposure quartile. While the CA : level 3 opioid exposure interaction is of borderline significance in all patients and women (p=0.0546 and 0.0568, but not in men) in the longitudinal study, the age : level 2 exposure is significant in all patients and men (p=0.0003 and 0.0110) and level 3 opioid exposure approaches significance in women as a factor and in interaction with age (p=0.0501 and 0.0547). Online supplementary table S9 presents the results of a longitudinal regression of age-corrected indices of cardiovascular stiffness against time and opioid exposure quartile by gender, and notes a highly significant effect by exposure quartile in both genders.

Corrected Indices by Time by Opioid Dependency Status

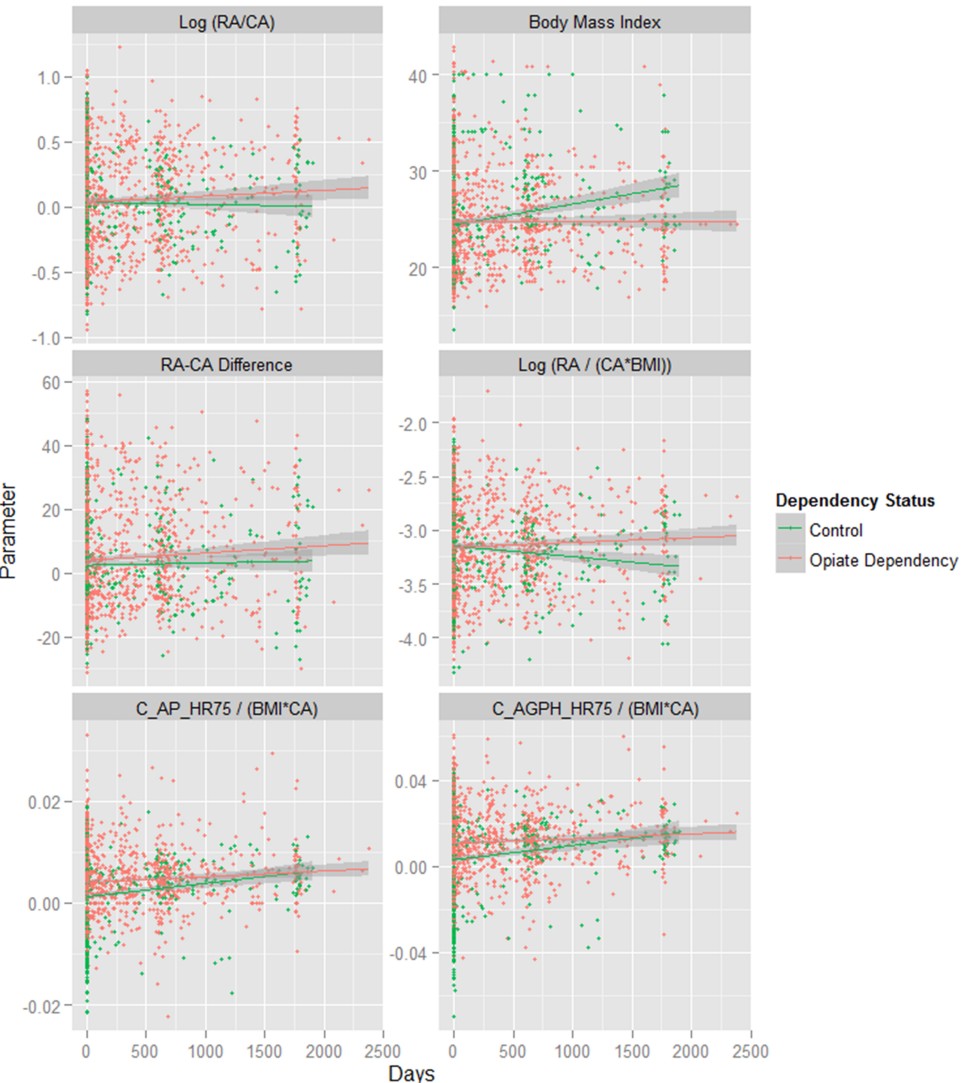

**Figure 2** Corrected ageing and stiffness indices by time by opioid dependency status. RA, Vascular Reference Age; CA, Chronological Age; C_AP_HR75, Central Augmentation Pressure at Heart Rate 75 bpm; C_AGPH_HR75, Central Augmentation Pressure/Pulse Height Ratio at Heart Rate 75 bpm.

The data were formally interrogated for a trend between the different quartiles of lifetime opioid exposure using the contrasts polynomial function in 'R'. Significant contrasts between the various quartiles for lifetime opioid exposure for the age : opioid exposure interaction was indeed demonstrated in all patients (est.=0.1657, SE=0.0787, t=2.104, p=0.0356; Model Adj. $R^2$=0.4265, F=133.0, df=7, 1235, p = 0.0001 such that moving up a quartile level at age 60 increased the modelled RA by 8.42% to 65.051 years, and movement from the first (non-exposed) quartile to the highest (level 3) quartile increased the modelled RA 25.26% to 75.154 years.

Some environmental exposure–pathology relationships have been shown to be power functions of the duration of exposure and are therefore particularly sensitive to the periods for which exposure occurs.[35] Since opioid administration is often prescribed and recommended as being of indefinite duration, this is particularly relevant to the present discussion. Therefore, a detailed study of the power relationships of the duration of opioid administration was undertaken. Table 3 shows that simple regression models of RA/CA against the dose–duration interaction were significantly related to linear, quadratic, cubic and quartic terms in opioid exposure duration. Moreover, when RA/CA was regressed against a model including interactive terms in linear, quadratic, cubic and quartic terms, all four were significant. When these models were compared by ANOVA studies, the quadratic model (AIC=82.888) was noted to be significantly better than the linear, cubic or quartic models (ANOVA tests, all p<0.05), but not significantly different from the combined model (AIC=83.731, p=0.87; online supplementary table S10).

When a similar exercise was performed in the longitudinal dataset, similar results were obtained (see online

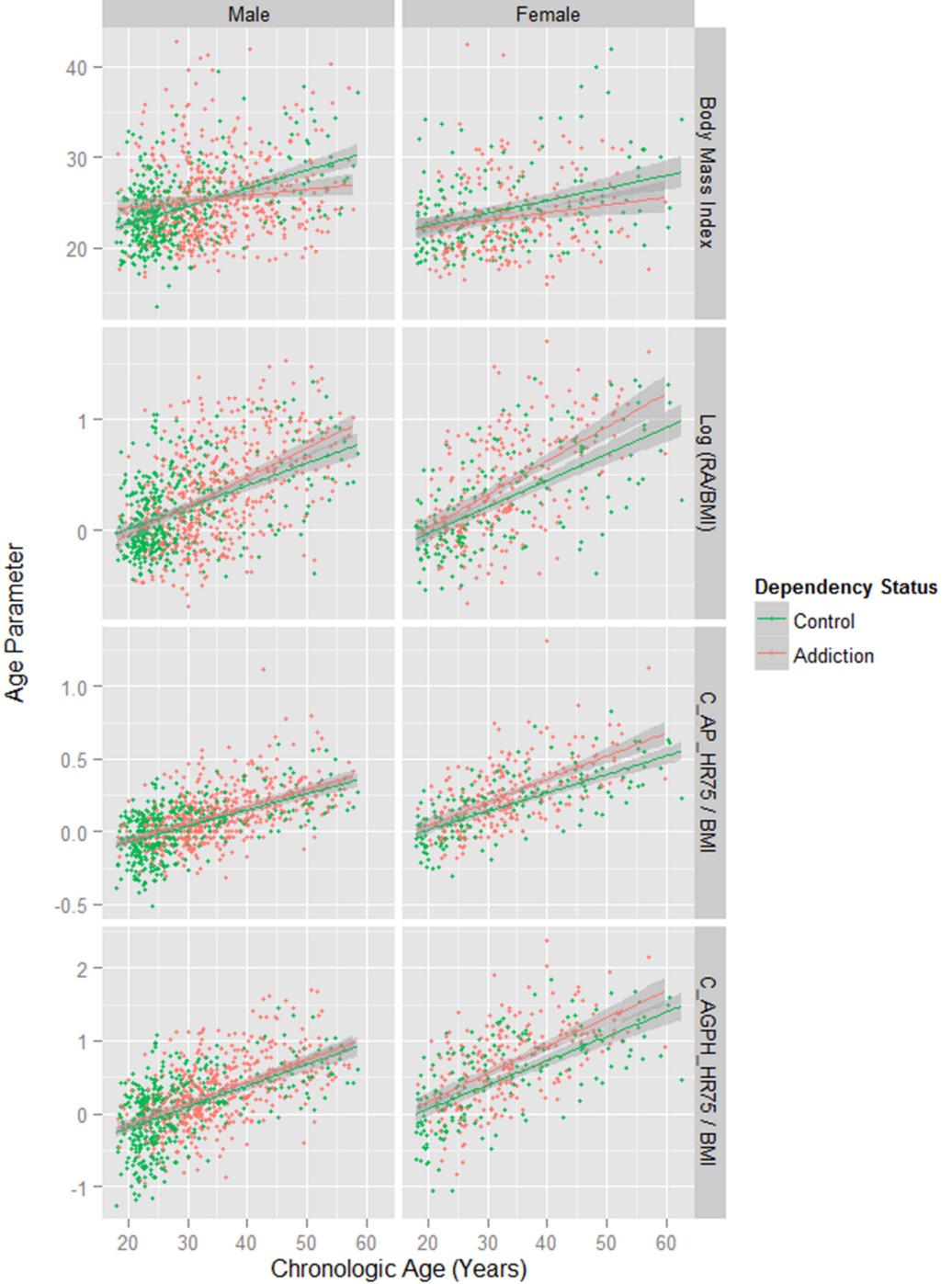

## BMI-Related Ageing Indices by Chronologic Age by Sex by Opioid Dependency Status

**Figure 3** Body mass index-related ageing indices by chronological age by addictive status by sex. RA, Vascular Reference Age; CA, Chronological Age; C_AP_HR75, Central Augmentation Pressure at Heart Rate 75 bpm; C_AGPH_HR75, Central Augmentation Pressure/Pulse Height Ratio at Heart Rate 75 bpm.

supplementary table S11). In this study RA/CA was regressed against time and the opioid exposure quartile. As a combined linear–quadratic–cubic–quartic model failed to converge, the highest order model used was linear–quadratic–quartic. Again, quadratic and quartic terms in exposure duration persist in the final model. In this case the quadratic model (AIC=1040.85) was significantly superior to all the others but was not significantly better than the linear–quadratic–quartic model (AIC=1041.09, p=0.082; see online supplementary table S12).

The data were interrogated for the presence of a dose–response relationship directly by examining continuous

## BMI-Related Ageing Indices by Chronologic Age by Sex by Opioid Dependency Status

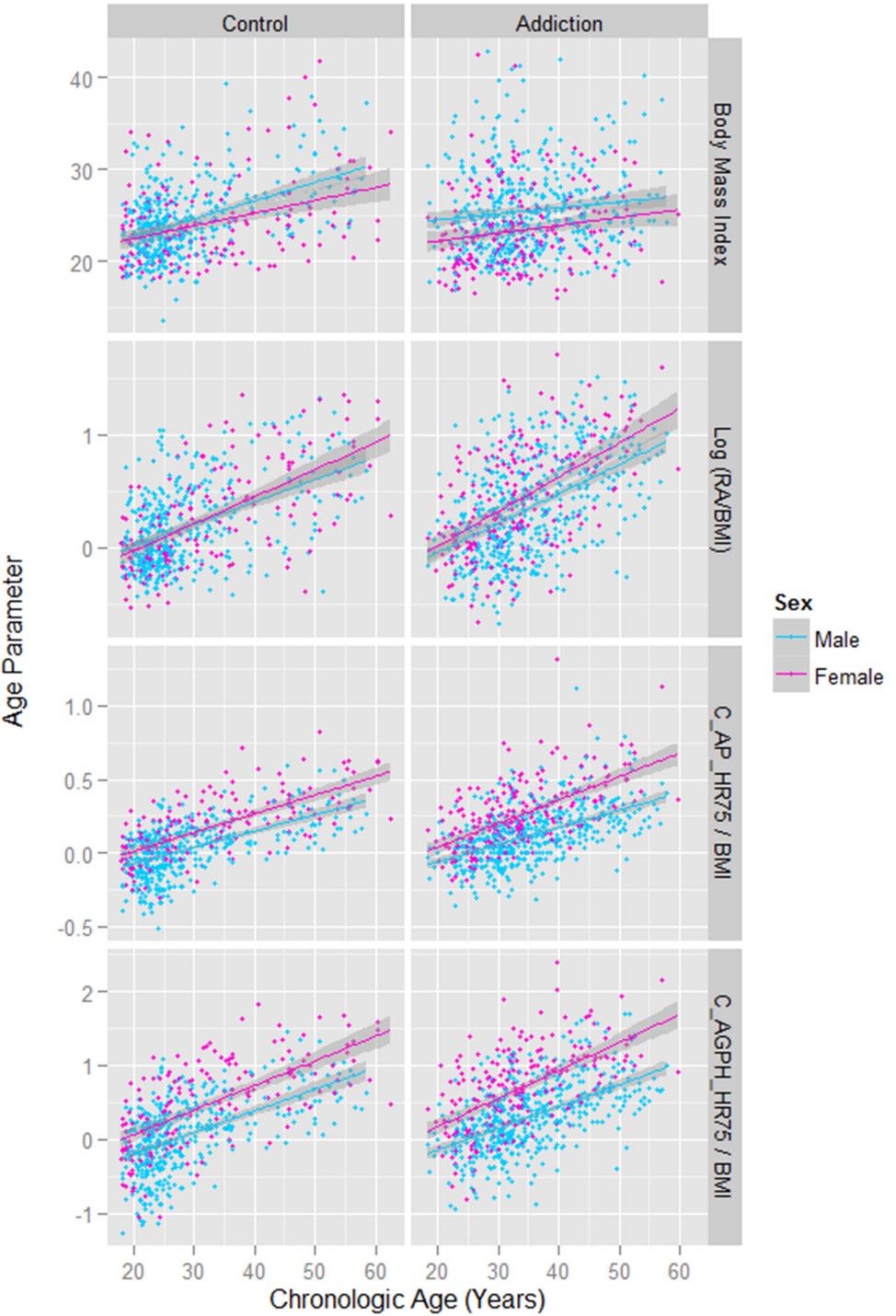

**Figure 4** Body mass index-related ageing indices by chronological age by sex by addictive status. RA, Vascular Reference Age; CA, Chronological Age; C_AP_HR75, Central Augmentation Pressure at Heart Rate 75 bpm; C_AGPH_HR75, Central Augmentation Pressure/Pulse Height Ratio at Heart Rate 75 bpm.

terms in opioid dose and duration regressed against the RA/CA ratio in both datasets. As shown in table 4 the dose–duration interaction is significant in all patients considered together, in men and in women ($p < 0.0001$, 0.0003 and 0.0236, respectively) in the cross-sectional dataset, while in the longitudinal dataset the parallel results are $p = 0.0004$, 0.0008 and 0.0127, respectively.

The final step of the analysis was to consider whether the significance of the opioid dose and duration might persist after adjustment for all known cardiovascular risk

**Table 1** Effects of gender on key measures of arterial stiffness—cross-sectional dataset

| Dependent variable | Group | Parameter | Adj. R² | F | Model P | df | Estimate | Upper limit | Lower limit | p Value |
|---|---|---|---|---|---|---|---|---|---|---|
| RA–CA difference | Opioid | Chron.Age:Female.Sex | 0.0189 | 7.596 | 0.0005 | 2, 684 | 0.6324 | 1.3123 | −0.0475 | 0.0688 |
| BMI | Control | Chron.Age:Female.Sex | 0.1523 | 52.58 | <0.0001 | 2, 572 | −0.0092 | −0.0014 | −0.0170 | 0.0213 |
| BMI | Opioid | Female.Sex | 0.0713 | 27.35 | <0.0001 | 2, 684 | −0.0827 | −0.0555 | −0.1099 | <0.0001 |
| RA/BMI | Opioid | Chron.Age:Female.Sex | 0.2617 | 122.6 | <0.0001 | 2, 684 | 0.0361 | 0.0539 | 0.0183 | <0.0001 |
| C_AP_HR75/BMI | Control | Chron.Age:Female.Sex | 0.4882 | 274.3 | <0.0001 | 2, 571 | 0.0312 | 0.0390 | 0.0234 | <0.0001 |
| C_AP_HR75/BMI | Control | Chron.Age:Female.Sex | 0.4882 | 274.3 | <0.0001 | 2, 571 | 0.0312 | 0.0390 | 0.0234 | <0.0001 |
| C_AP_HR75/BMI | Opioid | Chron.Age:Female.Sex | 0.3712 | 203.4 | <0.0001 | 2, 684 | 0.0467 | 0.0543 | 0.0391 | <0.0001 |
| C_AGPH_HR75/BMI | Control | Chron.Age:Female.Sex | 0.4603 | 245.4 | <0.0001 | 2, 571 | 0.0924 | 0.1144 | 0.0704 | <0.0001 |
| C_AGPH_HR75/BMI | Opioid | Chron.Age:Female.Sex | 0.3980 | 227.8 | <0.0001 | 2, 684 | 0.1265 | 0.1451 | 0.1079 | <0.0001 |
| C_AP_HR75/(CA*BMI) | Control | Female.Sex | 0.2809 | 112.9 | <0.0001 | 2, 571 | 0.0033 | 0.0043 | 0.0023 | <0.0001 |
| C_AP_HR75/(CA*BMI) | Opioid | Female.Sex | 0.2489 | 114.7 | <0.0001 | 2, 684 | 0.0047 | 0.0055 | 0.0039 | <0.0001 |
| C_AGPH_HR75/(CA*BMI) | Control | Female.Sex | 0.2567 | 99.96 | <0.0001 | 2, 571 | 0.0096 | 0.0125 | 0.0067 | <0.0001 |
| C_AGPH_HR75/(CA*BMI) | Opioid | Female.Sex | 0.2604 | 121.8 | <0.0001 | 2, 684 | 0.0130 | 0.0152 | 0.0108 | <0.0001 |

factors. In preliminary analyses lipidic factors were invariably eliminated during the model reduction process. Moreover, as discussed in the Introduction, opioids themselves are likely to perturb lipid function, and also have a robust immunostimulatory action. Therefore, including immune markers is likely to over-correct the model. Similarly, when time since last cigarette consumption was included in the cross-sectional and longitudinal models it was eliminated during model reduction. For these reasons the final cross-sectional formula used regressed RA/CA against interactive terms in opioid dose and duration, height, tobacco consumption, systolic blood pressure and weight. The longitudinal model was similar but had an additive term in weight. These were the terms that remained significant in the models in preliminary exploratory analyses for log (RA). Further consideration of the clinical justification for the choice of this five-way interaction is included in the Discussion section.

The results shown in table 5 were found for the cross-sectional data. This table shows that significant interactions including opioid dose–duration terms are found in all patients (p=0.0261) and in men (p=0.0017; Adj. R²=0.0815), while in women this interaction approaches significance (p=0.0516). When a similar exercise is performed on the longitudinal data, significant interactions are seen from p=0.0011, 0.0013 and 0.0073 and above in each group, respectively (table 6).

Online supplementary table S13 presents the results of final models in which complete multiple regression was performed in the cross-sectional dataset for all pertinent variables. (Log) RA/CA was regressed against interactive terms in opioid dose and duration, height, tobacco use, systolic pressure and high sensitivity CRP together with additive terms in weight, cholesterol and HDL. Opioid dose and duration persist in the final model in men and in all patients, but not in women. Interestingly CRP features in most of the terms of these final models.

These overall results were broadly confirmed by the running of further analyses including by the use of additivity and variance stabilisation in R.[36] In particular, we analysed a range of models including different polynomial terms in known predictors, and matching models augmenting the known risk factors of heroin use. Models with and without heroin use were compared. These analyses confirmed that the model fit for the relationship between log (RA/CA) and the predictive variables was significantly improved by the inclusion of terms for opiate exposure (dose and duration) among the male opiate dependents, in the opioid-dependent groups in both sexes and in all patients; that the relationship between the various model parameters was (unsurprisingly) unlikely to be of a strictly linear form and that superior model fits for log (RA/CA) were likely to be accompanied by various higher order interactions between the most powerful predictive independent variables, namely, (log) SP, weight, height, tobacco consumption and opioid exposure dose and duration.

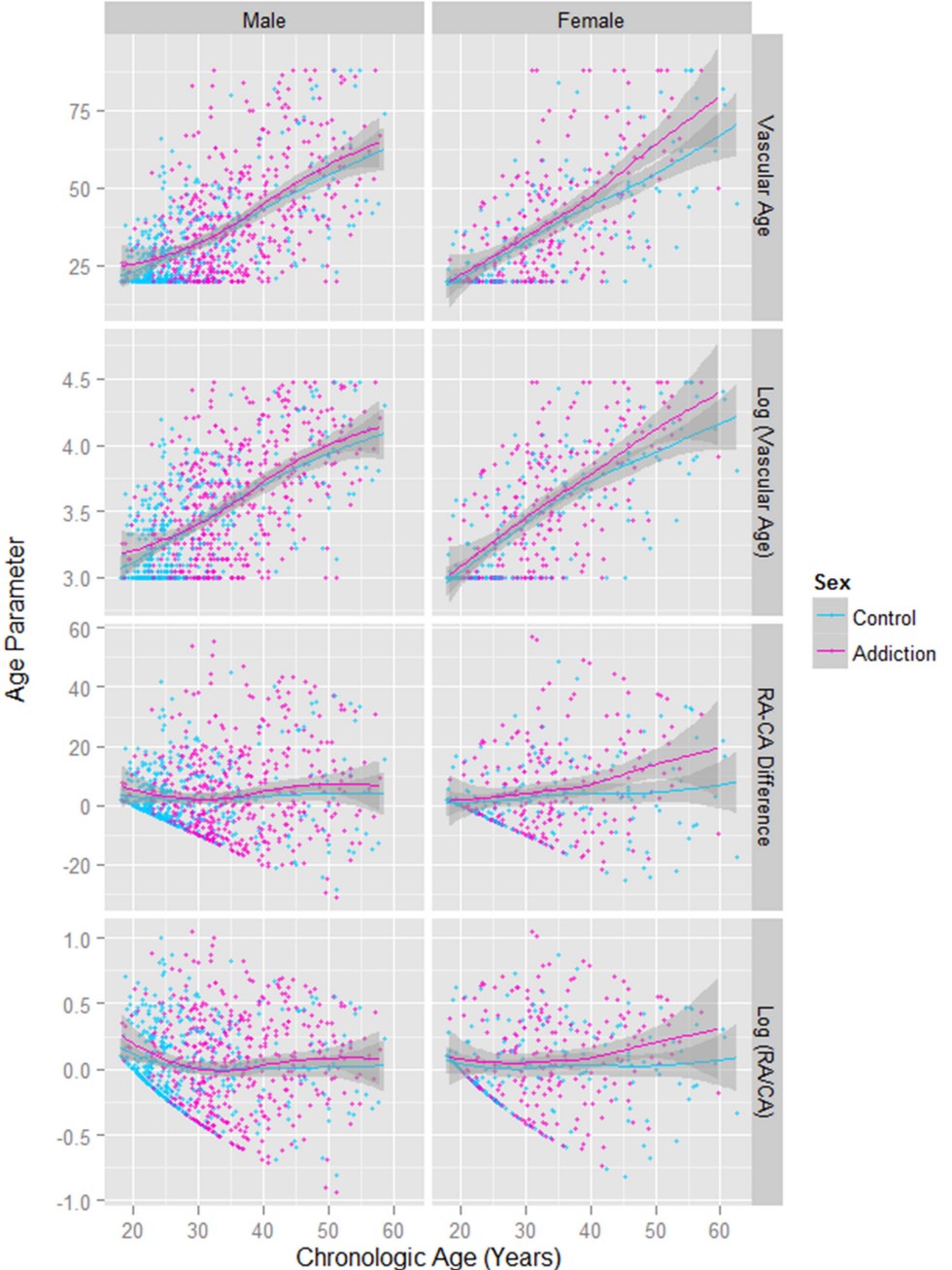

**Figure 5**  Ageing indices by chronological age by sex by addictive status—LOESS curves. RA, Vascular Reference Age; CA, Chronological Age.

## DISCUSSION

The major findings of this study are that indices of vascular age and arterial stiffness are worse in opioid-dependent patients compared with opioid naïve controls with mean calculated ages elevated by 1.97% in men and 13.43% in women. The effect was thus more marked in women. A significant effect was found on vascular age and augmentation index by exposure quartile after correction for CA and BMI. The RA/CA ratio was found to be related to power functions of the opioid

duration of exposure in cross-sectional and longitudinal analyses. A dose–response relationship was demonstrated with lifetime opioid exposure. In particular, the effect of opioid exposure was robust, and remained after multiple adjustments in cross-sectional and longitudinal studies.

These findings should be interpreted in the light of the relatively modest degree of opioid exposure to which these patients were exposed. While the dose and duration of opioids used by patients in this study is typical of that seen in many published series, the treatment of

**Table 2** Age: addictive status relationships with CA discretised, about 41 years in men and 29 years in women with women <55 years of age

| Parameter | Variable statistics | | | | Model summaries | | |
|---|---|---|---|---|---|---|---|
| | Estimate | Upper limit | Lower limit | p Value | F | df | p Value |
| Males | | | | | | | |
| Older age | −0.1837 | −0.0745 | −0.2929 | 0.0010 | 5.492 | 3, 862 | 0.0010 |
| Opioids | −0.0846 | −0.0307 | −0.1385 | 0.0021 | | | |
| Older age:Opioids | 0.2310 | 0.3657 | 0.0963 | 0.0008 | | | |
| Females | | | | | | | |
| Older age:Opioids | 0.1248 | 0.2291 | 0.0205 | 0.0195 | 2.169 | 3, 378 | 0.0912 |

these patients was mostly with the relatively mild partial agonist buprenorphine, with only modest doses employed (mean 6.98±0.21 mg): this dose level being much lower than the commonly recommended (18–24 mg).[37] Moreover, buprenorphine has been shown to be more mild in its cardiovascular effects than the full agonist agent methadone . For this reason the results reported here should be seen as a lower bound on the effects which may be observed.

The findings in relation to central vascular stiffness and age being related to power functions of the treatment duration are of great importance. Chronic opioid agonist treatment is often seen as relatively benign, and partly for this reason, is frequently recommended for very long term or indefinite administration.[38] [39] However, the demonstration here that the effects likely compound with time has major clinical practice implications, and has not been previously noted. The 1.97%

effect described here for men, compounded over 30 years equates cumulatively to 179.5% gain and in diminution to 44.95% loss. Similarly the 13.43% gain seen in women compounded over 30 years is a 4349.02% gain and a 98.66% loss in reduction. The effects, therefore, of these processes maintained over the long term on stem cell diminution or immune stimulation are potentially very considerable.

The exacerbation of the effect in women was clearly demonstrated in figures 3–5, tables 1 and 2, online supplementary tables 4 and 5 and online supplementary figures 1 and 2. This finding has, to the authors' knowledge, only been reported previously by an Iranian group.[14] If one looks closely at figure 5, several fascinating findings emerge. The vascular age algorithm used by the SphygmoCor software is based on the augmentation index C_AGPH_HR75. This index has a clear curvilinear convex upward relationship with CA as shown in figure 5

**Table 3** Cross-sectional models polynomial in heroin duration

| Model/parameter | Parameter | | | | Model | | |
|---|---|---|---|---|---|---|---|
| | Value | Upper limit | Lower limit | p Value | F | df | p Value |
| *1 Order—Linear* | | | | | | | |
| H.Dura.n | −0.0059 | −0.0018 | −0.0100 | 0.0056 | 8.628 | 2, 664 | 0.0002 |
| H.Dose: H.Dura.n | 0.0091 | 0.0134 | 0.0048 | 3.7.E−05 | | | |
| *2 Order—Quadratic* | | | | | | | |
| H.Dura.n | −0.0171 | −0.0063 | −0.0279 | 0.0018 | 7.443 | 3, 663 | <0.0001 |
| H.Dura.n^2 | 0.0004 | 0.0008 | 0.0000 | 0.0261 | | | |
| H.Dose: H.Dura.n | 0.0084 | 0.0127 | 0.0041 | 0.0002 | | | |
| *3 Order—Cubic* | | | | | | | |
| H.Dura.n | −0.0116 | −0.0047 | −0.0185 | 0.0012 | 7.102 | 3, 663 | 0.0001 |
| H.Dura.n^3 | 6.3E−06 | 0.0000 | 0.0000 | 0.0466 | | | |
| H.Dose: H.Dura.n | 0.0084 | 0.0127 | 0.0041 | 0.0001 | | | |
| *4 Order—Quartic* | | | | | | | |
| H.Dura.n | −0.0059 | −0.0018 | −0.0100 | 0.0056 | 8.628 | 2, 664 | 0.0002 |
| H.Dose: H.Dura.n | 0.0091 | 0.0134 | 0.0048 | 3.7.E−05 | | | |
| *1, 2, 3 & 4 Order* | | | | | | | |
| H.Dose: H.Dura.n^3 | 6.5E−05 | 1.0E−04 | 2.6E−05 | 0.0012 | 3.353 | 7, 659 | 0.0016 |
| H.Dura.n^2 : H.Dura.n^3 | −1.0E−06 | −3.9E−07 | −1.6E−06 | 0.0007 | | | |
| H.Dura.n^2 : H.Dura.n^4 | 9.2E−08 | 1.5E−07 | 3.5E−08 | 0.0014 | | | |
| H.Dura.n^3: H.Dura.n^4 | −2.7E−09 | −9.8E−10 | −4.4E−09 | 0.0026 | | | |
| H.Dose: H.Dura.n^2 : H.Dura.n^4 | −4.5E−09 | −9.7E−10 | −8.0E−09 | 0.0145 | | | |
| H.Dura.n: H.Dura.n^3: H.Dura.n^4 | 2.5E−11 | 4.3E−11 | 7.4E−12 | 0.0048 | | | |
| H.Dose: H.Dura.n: H.Dura.n^3: H.Dura.n^4 | 2.6E−12 | 5.0E−12 | 2.5E−13 | 0.0312 | | | |

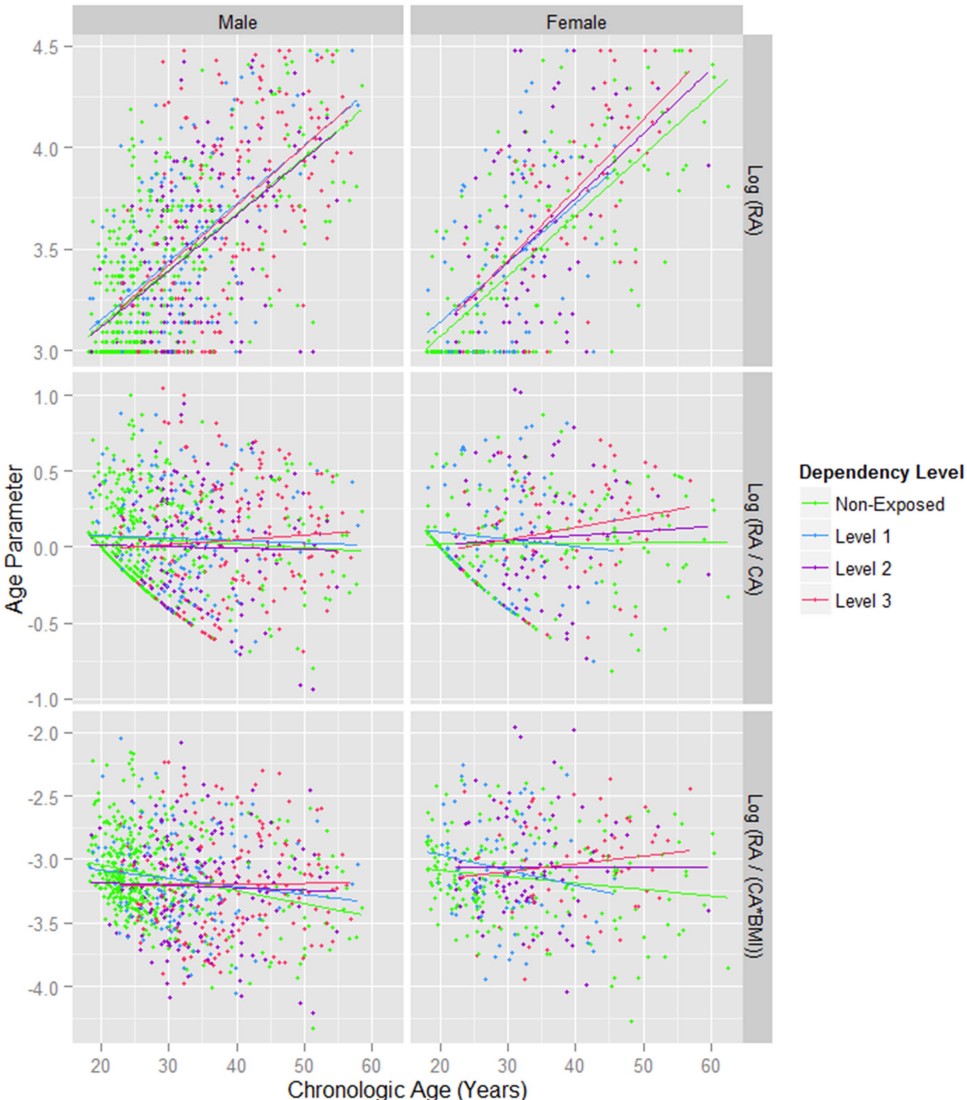

**Figure 6** Vascular age indices by chronological age by opioid dependency quartile. RA, Vascular Reference Age; CA, Chronological Age. Opioid exposure quartiles as defined in the Methods section.

in the lower panels, unlike the C_AP_HR75 which is directly linear.[40] Hence the reduction at higher ages on the C_AGPH_HR75 curve may be somewhat artefactual. If one looks at the upper panels of figure 5 the rise in RA with age in ODP may be seen to be broadly exponential in form, if the decline in slope above the age of 45 years is discounted. In women it appears that the rise in RA occurs some 10 years prior to that seen in men, a finding emphasised by the middle panels. This fits with a dramatic clinical transformation which is seen in opioid-dependent women at around the age of 30, and in men at around the age of 40. While the cause of this is unknown it may possibly be related to a more aggressive immune system in women which is in part hormonally mediated.[41] In this context the demonstration that oestradiol directly stimulates telomerase was particularly provocative.[42]

Some consideration of the comparability of the control and opioid-dependent groups is appropriate. Significant differences were found between the control and opioid-dependent groups in relation to CA, weight, BMI, smoking, cholesterol, low-density lipoprotein (LDL) and hepatic and inflammatory laboratory markers, but these were controlled for by various multiple regression techniques. The higher mean weight in the opioid-dependent group may have been related in part to their higher age, and in part to the orexigenic effects of opioid agonism[43] while serial resource diversion may explain the slower rate of gain of weight in this group over time. Higher rates of tobacco consumption may be related to either the common features of the addictive lifestyle or the proconsumptive effects of opioid analogues.[44] Significantly more of the women in the control group (11.4% vs 5.41%) were aged over

**Table 4** Dose–response relationships

| Parameters | Variable statistics | | | | Model summaries | | |
|---|---|---|---|---|---|---|---|
| | Estimate | Upper limit | Lower limit | Pr(>ltl) | F | df | p Value |
| Cross-sectional studies | | | | | | | |
| *Both Sexes* | | | | | | | |
| H.Dura.n | −0.0059 | −0.0018 | −0.0100 | 0.0056 | 8.628 | 2, 664 | 0.0002 |
| H.Dose: H.Dura.n | 0.0091 | 0.0134 | 0.0048 | 0.0000 | | | |
| *Males* | | | | | | | |
| H.Dura.n | −0.0089 | −0.0034 | −0.0144 | 0.0018 | 7.105 | 2, 441 | 0.0009 |
| H.Dose: H.Dura.n | 0.0124 | 0.0191 | 0.0057 | 0.0003 | | | |
| *Females* | | | | | | | |
| H.Dose: H.Dura.n | 0.0052 | 0.0097 | 0.0007 | 0.0236 | 5.201 | 1, 210 | 0.0236 |

| Group | Variable statistics | | | | | Model | |
|---|---|---|---|---|---|---|---|
| Parameter | Value | Upper limit | Lower limit | t-value | p Value | AIC* | LogLik* |
| Longitudinal studies | | | | | | | |
| *Both sexes (df=814)* | | | | | | | |
| H.Dose:H.Dura.n | 0.0045 | 0.0070 | 0.0020 | 3.5512 | 0.0004 | 1068.74 | −530.37 |
| *Males (df=552)* | | | | | | | |
| H.Dura.n | −0.0036 | −0.0005 | −0.0067 | −2.2069 | 0.0277 | 753.15 | −371.58 |
| H.Dose:H.Dura.n | 0.0075 | 0.0118 | 0.0032 | 3.3754 | 0.0008 | | |
| *Females (df=260)* | | | | | | | |
| H.Dose:H.Dura.n | 0.0052 | 0.0093 | 0.0011 | 2.5091 | 0.0127 | 347.43 | −169.72 |

50 years (p=0.029). However as no data were collected relating to hormonal status with regard to the menopause, it is not possible to remark on the relative hormonal status of the two groups. Opioid dependency is well known to be associated with hypothalamic hypogonadotrophic oligoamenorrhoea, so that age-related hormonal suppression in the control group was somewhat balanced by drug-related hormonal suppression in the opioid-dependent group. Clarification of this issue must await further studies.

Some consideration of the clinical justification for the five-way interaction is appropriate which was presented in the final multiple regression models of RA (tables 5 and 6). Height is a key variable that relates to many techniques of measuring of arterial stiffness since it is related to the timing of arterial wave reflections, and it is hardly surprising that this remains in the final model.[45 46] Height and weight will naturally interact and indeed the BMI is one formula which formalises such interactions. The opioid dose–duration exposure is a standard way in which to account for lifetime opioid exposure, and has been used by WHO[35] and NIH[14] investigators among others. Other factors including laboratory parameters were not present in the final models. Therefore, it is not surprising that this cluster of factors remains in the final model.

On the other hand, CRP was prominent and present in most terms in the final model when RA/CA was interrogated by multiple regression (see online supplementary table S13) suggesting that innate immune processes may be implicated in the observed opioid-vasculopathy effect. The age-related course of high-sensitivity CRP has been described in several earlier reports in our patient cohort.[6 28 47] Online supplementary table S1 shows significant elevation of CRP, erythrocyte sedimentation rate, serum globulins and circulating lymphocytes and monocyte numbers in the opioid-dependent group. Consideration of the very flat age-dependent trajectories of these parameters in controls in our earlier reports shows that this effect is much greater in opioid dependence than might be expected on the basis of the higher CA in the opioid-dependent group alone. This is in turn consistent with the important chronic inflammatory basis of many vascular diseases[48–50] and the immune theory of organismal ageing.[48 51]

As noted above, vascular age is frequently said to be a surrogate for organismal ageing. The present data are consistent with that view, and also with the literature cited of accelerated ageing in these patients. The present technique provides a mechanism to quantify these changes at the whole human level. Interestingly, while it is well known that telomere length can function as a 'mitotic clock' of the biological ageing process, it has been shown that oxidative stress can stochastically increase this rate, in part by inducing single-strand telomeric breaks related to the high guanosine content of telomeric repeats $(TTAGGG)n$.[52–54] The heightened inflammatory activity associated with opioid dependence has been previously noted,[5–7 28 55] and the involvement of this pathway mechanistically unifies the inflammatory, oxidative, chromosomal, telomeric and stem cell hypotheses of ageing, and accounts for the uniformity of the progeroid findings across organ systems[56] and the high prevalence of multisystem disease seen in opioid dependence.[11 12 28]

As no structural imaging techniques were performed, it is not possible to say with certainty if the observed

**Table 5** Cross-sectional adjusted regression models

| Group | Variable statistics | | | | Model | | |
|---|---|---|---|---|---|---|---|
| Parameter | Estimate | Upper limit | Lower limit | Pr(>|t|) | F | df | p Value |
| *Both sexes* | | | | | | | |
| H.Dura.n: Cigs.d: SP : Weight | 0.00023 | 0.0003 | 0.0001 | 0.0001 | 7.561 | 9, 657 | <0.0001 |
| H.Dura.n: Height: Cigs.d: SP | 0.00030 | 0.0005 | 0.0001 | 0.0047 | | | |
| H.Dura.n: Height: Cigs.d | −0.00143 | −0.0004 | −0.0024 | 0.0048 | | | |
| H.Dura.n: Height: Cigs.d: SP : Weight | −0.00007 | 0.0000 | −0.0001 | 0.0051 | | | |
| H.Dura.n: Height: Cigs.d: Weight | 0.00032 | 0.0006 | 0.0001 | 0.0063 | | | |
| H.Dose: H.Dura.n: Height: Cigs.d: Weight | −0.00037 | −0.0001 | −0.0007 | 0.0261 | | | |
| H.Dose: H.Dura.n: Height: Cigs.d: SP : Weight | 0.00008 | 0.0001 | 0.0000 | 0.0262 | | | |
| H.Dose: H.Dura.n: Height: Cigs.d | 0.00155 | 0.0029 | 0.0002 | 0.0286 | | | |
| H.Dose: H.Dura.n: Height: Cigs.d: SP | −0.00032 | 0.0000 | −0.0006 | 0.0290 | | | |
| *Males* | | | | | | | |
| H.Dura.n: Weight | −0.0024 | −0.0010 | −0.0038 | 0.0007 | 5.485 | 10, 438 | <0.0001 |
| H.Dose: H.Dura.n: Weight | 0.0032 | 0.0052 | 0.0012 | 0.0017 | | | |
| H.Dose: Cigs.d: SP : Weight | −4.4770 | −1.0529 | −7.9011 | 0.0107 | | | |
| H.Dose: Height: Cigs.d: SP : Weight | 0.0253 | 0.0447 | 0.0059 | 0.0109 | | | |
| H.Dose: Cigs.d: SP | 19.460 | 34.3874 | 4.5326 | 0.0109 | | | |
| H.Dose: Cigs.d: Weight | 21.370 | 37.8144 | 4.9256 | 0.0112 | | | |
| H.Dose: Height: Cigs.d: Weight | −0.1209 | −0.0276 | −0.2142 | 0.0114 | | | |
| H.Dose: Height: Cigs.d: SP | −0.1098 | −0.0251 | −0.1945 | 0.0114 | | | |
| H.Dose: Cigs.d | −92.680 | −21.100 | −164.25 | 0.0115 | | | |
| H.Dose: Height: Cigs.d | 0.5230 | 0.9293 | 0.1167 | 0.0120 | | | |
| *Females* | | | | | | | |
| Height: Cigs.d: Weight | −0.00041 | −0.0002 | −0.0006 | 0.00002 | 8.25 | 3, 214 | <0.0001 |
| Height: Cigs.d: SP : Weight | 0.00008 | 0.0001 | 0.0000 | 0.00003 | | | |
| H.Dose: H.Dura.n: Height: Cigs.d | 2.2E−06 | 0.0000 | 0.0000 | 0.05160 | | | |

elevations in vascular stiffness were associated with fixed atherosclerotic lesions. However a large and weighty literature confirms the relevance of significance of elevations in vascular stiffness.[45 46 57–59] Structural correlation must await further work.

The consistency of the present results with the previously reported data from various national and international groups[11–18 60] is clear in that we have confirmed the exacerbation of cardiovascular ageing and arterial stiffness in opioid dependency. However these earlier results have been extended by demonstrating a measurable major subclinical endophenotype of accelerated ageing antemortem, confirming dose–response relationships, demonstrating for the first time the significance of the improved quadratic model in opioid exposure duration and showing that the opioid-related effect is robust to adjustment for known cardiovascular risk factors in both sexes. Moreover, we confirmed that the effect is worse in women[14] and extended this result by showing that the differential effect starts in women over a decade earlier than in men (29 vs 41 years). In that the PWA technique directly relates increased arterial stiffness to cardiovascular and therefore, organismal ageing, the present detailed studies link advanced cardiovascular age to many other indices of accelerated ageing in opioid dependency as

have been shown by this[6 7 10 26 28 47 61–65] and other[66] groups. As such the present report is a meaningful and important extension of earlier reports from this project on the cardiovascular implications of opioid dependency,[30–33] its treatment[19] and withdrawal[23] and the antagonist-induced reversal of these effects.[20] Hence the phenomenology of the present descriptive report is consistent with, and extends, extant reports in the published literature.

While this study is not designed to be mechanistic in nature, some brief comment on this issue may be appropriate, in part for its relevance to the consideration of possibly aetiologically causal pathways. Some of the most interesting questions to emerge from the present work relate to the possible mechanisms underlying the plethora of changes which have been identified in opioid dependence as cardiovascular risk factors, as potentially atherogenic processes and as accelerants of ageing processes. Opioids have been linked with blockade of the cell cycle at cyclin dependent kinases 2, 4 and 6 by P16$^{INK4A}$ and P21$^{WAF1/CIP1}$.[67 68] P16 is the gene product of CDKN2A located at the senescence locus on chromosome 9p21.3 which is a hotspot for many Genome Wide Association Studies (GWAS) studies, and from which three senescence-related proteins and one senescenc-associated long non-coding RNA are

**Table 6** Longitudinal adjusted regression models

| Group | Variable statistics | | | | | Model | |
|---|---|---|---|---|---|---|---|
| Model/Parameter | Value | Upper limit | Lower limit | t Value | p Value | AIC* | LogLik* |
| *Both sexes (df=606)* | | | | | | | |
| Height | −0.0304 | −0.0253 | −0.0355 | −11.7198 | 0.0E+00 | 1238.239 | −592.1197 |
| SP | 0.7947 | 0.9807 | 0.6087 | 8.3772 | 0.0E+00 | | |
| Days:H.Dura.n | 0.0003 | 0.0004 | 0.0002 | 4.3556 | 0.0E+00 | | |
| CA:Height | 0.0066 | 0.0078 | 0.0054 | 10.9049 | 0.0E+00 | | |
| Days: CA:H.Dura.n | −0.0001 | −0.0001 | −0.0001 | −4.2946 | 0.0E+00 | | |
| H.Dura.n: Cigs.d:Height | −5.3E−05 | −2.8E−05 | −7.8E−05 | −4.1566 | 0.0E+00 | | |
| Days:H.Dose: Cigs.d:Height | 2.1E−06 | 3.1E−06 | 1.1E−06 | 4.3108 | 0.0E+00 | | |
| Days: CA:H.Dose: Cigs.d:Height | −6.0E−07 | −2.1E−07 | −9.9E−07 | −4.2968 | 0.0E+00 | | |
| CA:H.Dura.n: Cigs.d:Height | 1.4E−05 | 2.1E−05 | 7.3E−06 | 4.0217 | 0.0001 | | |
| Days: CA: Cigs.d:Height | 0.0E+00 | 0.0E+00 | 0.0E+00 | 3.7701 | 0.0002 | | |
| Days:H.Dura.n: Cigs.d | −1.1E−05 | −4.9E−06 | −1.7E−05 | −3.4709 | 0.0006 | | |
| Cigs.d:Height | 0.0007 | 0.0011 | 0.0003 | 3.3714 | 0.0008 | | |
| Days: CA:H.Dose:H.Dura.n: Cigs.d: Height | 0.0E+00 | 0.0E+00 | 0.0E+00 | 3.2705 | 0.0011 | | |
| Days: CA:H.Dura.n: Cigs.d | 2.5E−06 | 4.1E−06 | 9.3E−07 | 3.2365 | 0.0013 | | |
| CA: Cigs.d:Height | −0.0002 | 0.0000 | −0.0004 | −3.2045 | 0.0014 | | |
| Days | −0.0022 | −0.0008 | −0.0036 | −2.9194 | 0.0036 | | |
| H.Dose:H.Dura.n: Cigs.d | 0.0102 | 0.0173 | 0.0031 | 2.8304 | 0.0048 | | |
| CA:H.Dose:H.Dura.n: Cigs.d | −0.0026 | −0.0006 | −0.0046 | −2.6804 | 0.0076 | | |
| Days: CA | 0.0005 | 0.0009 | 0.0001 | 2.2438 | 0.0252 | | |
| Days: CA:H.Dose:Height | 6.8E−06 | 1.3E−05 | 7.2E−07 | 2.1984 | 0.0283 | | |
| Days:H.Dose:Height | −2.2E−05 | −2.4E−06 | −4.2E−05 | −2.1747 | 0.0300 | | |
| Days:H.Dose:H.Dura.n | −3.5E−05 | −1.7E−06 | −6.8E−05 | −2.1086 | 0.0354 | | |
| H.Dose: Cigs.d | −0.1067 | −0.0065 | −0.2069 | −2.0888 | 0.0371 | | |
| CA:H.Dose: Cigs.d | 0.0290 | 0.0580 | 0.0000 | 1.9645 | 0.0499 | | |
| *Males (df=436)* | | | | | | | |
| Height | −0.0302 | −0.0233 | −0.0371 | −8.7423 | 0.0E+00 | 913.00 | −435.50 |
| SP | 0.6376 | 0.8542 | 0.4210 | 5.7717 | 0.0E+00 | | |
| CA:Height | 0.0064 | 0.0080 | 0.0048 | 8.4711 | 0.0E+00 | | |
| Days:H.Dura.n | 0.0003 | 0.0005 | 0.0001 | 3.8907 | 0.0001 | | |
| Days: CA:H.Dura.n | −0.0001 | −0.0001 | −0.0001 | −3.8395 | 0.0001 | | |
| Days:H.Dose: Cigs.d:Height | 1.6E−06 | 2.4E−06 | 8.2E−07 | 3.8181 | 0.0002 | | |
| Days: CA:H.Dose: Cigs.d:Height | −5.0E−07 | −3.0E−07 | −7.0E−07 | −3.7217 | 0.0002 | | |
| Days | −0.0029 | −0.0013 | −0.0045 | −3.6624 | 0.0003 | | |
| Days: CA | 0.0007 | 0.0011 | 0.0003 | 3.3251 | 0.0010 | | |
| H.Dose:H.Dura.n: Cigs.d:Height | 2.7E−06 | 4.3E−06 | 1.1E−06 | 3.2331 | 0.0013 | | |
| Days: CA: Cigs.d | 5.0E−06 | 8.1E−06 | 1.9E−06 | 3.1839 | 0.0016 | | |
| H.Dura.n: Cigs.d:Height | −2.8E−05 | −1.0E−05 | −4.6E−05 | −3.1058 | 0.0020 | | |
| CA:H.Dura.n: Cigs.d:Height | 7.1E−06 | 1.2E−05 | 2.4E−06 | 2.9496 | 0.0034 | | |
| Days:H.Dura.n: Cigs.d | −8.3E−06 | −2.2E−06 | −1.4E−05 | −2.6824 | 0.0076 | | |
| Days: CA:H.Dura.n: Cigs.d | 2.0E−06 | 3.6E−06 | 4.3E−07 | 2.5562 | 0.0109 | | |
| Days: CA:H.Dose:H.Dura.n: Cigs.d: Height | 0.0E+00 | 0.0E+00 | 0.0E+00 | 2.2838 | 0.0229 | | |
| Cigs.d:Height | 0.0004 | 0.0008 | 0.0000 | 2.2799 | 0.0231 | | |
| CA: Cigs.d:Height | −0.0001 | −2.0E−06 | −2.0E−04 | −2.1938 | 0.0288 | | |
| *Females (df=169)* | | | | | | | |
| Height | −0.0279 | −0.0208 | −0.0350 | −7.6753 | 0.0E+00 | 394.69 | −187.35 |
| SP | 1.2880 | 1.6663 | 0.9097 | 6.6744 | 0.0E+00 | | |
| CA:Height | 0.0064 | 0.0074 | 0.0054 | 11.7493 | 0.0E+00 | | |
| Days: CA:H.Dose:H.Dura.n:Height | 1.3E−06 | 2.3E−06 | 3.2E−07 | 2.7371 | 0.0069 | | |
| Days:H.Dose:H.Dura.n:Height | −4.5E−06 | −1.2E−06 | −7.8E−06 | −2.7147 | 0.0073 | | |
| Days: CA:H.Dose:H.Dura.n: Cigs.d: Height | −1.0E−07 | −1.0E−07 | −1.0E−07 | −2.6052 | 0.0100 | | |
| Days:H.Dose:H.Dura.n: Cigs.d:Height | 2.0E−07 | 4.0E−07 | 4.0E−09 | 2.5589 | 0.0114 | | |

transcribed.[69] Furthermore, opioid agonists have been shown to directly ligate the myeloid differentiation factor 2 (MD2)-toll-like receptor (TLR) 4 heterodimer leading to the downstream activation of the numerous powerful pathways linked to stimulation of the endotoxin receptor.[70] Indeed systemic endotoxaemia was recently shown to modulate the expression of 6025 genes.[71] Some of the pathways activated downstream by TLR4 signalling include the cyclooxygenase, lipooxygenase, MAP kinase (JNK and P38), AP-1, inducible nitric oxide synthase (iNOS), phosphoinositol-3 kinase (PI3K) and TGFβ pathways[72] many of which have been noted to be stimulated within the first few minutes of TLR4 ligation[73] to be stimulated by opioids[70 72] and to be associated with potentiating the effects of ageing.[51 74] Many of these immune products directly exacerbate the stem cell blockade.[75] This then implies that opioid dependency may be conceptualised as chronically, if largely subclinically, ill.

It is very exciting that new research has identified that these processes also act particularly in proopioimelano-cortin (POMC) neurons of the paraventricular and arcuate nuclei of the mediobasal hypothalamus which are key determinants of the body's metabolic homeostasis, glucostatic and lipostatic functions, energy balance, appetitive drives, sympathoadrenal activation state and even lifespan modulation in some organisms,[76–78] and that derangements in this region cause metabolic syndrome and its numerous related disorders in model organisms including mammals and likely also humans.[79] These workers have shown that the activation of the MD2/TLR4/IKKβ/NF-κB cascade induces endoplasmic reticulum, mitochondrial and autophagic stress, which along with reduced neurogenesis, increased POMC neuronal apoptosis[77] and glial activation and reactive gliosis[79 80] underlie these numerous disorders and derangements. While this research focuses on TLR4 ligation of unsaturated fatty acids and oxidised LDLs, opioids have been shown to act similarly as xenobiotics through MD2-TLR4[81] and opioid agonists have been shown to produce POMC cell hyperpolarisation and relative refractoriness[44] while opioid antagonists have the reverse effect.[82] Small molecule TLR4 inhibitors are under development for several therapeutic applications.[83 84] The confluence then of these two important lines of recent research has the potential to produce major conceptual and therapeutic advances in this field. A clear understanding of direct links from xenobiotic-TLR4-NF-κB signalling to sympathoadrenal tone and hypothalamic lifespan control has the potential to revolutionise the direction of therapeutic endeavours in the treatment of opioid dependence.

It is fascinating that the promoters for two of the key stem cell genes, Oct4 and Nanog, have response elements in their promoters for STAT-1 and STAT-3 which are commonly produced as a result of cytokine signalling, particularly by gp-130 cytokines such as IL-6 and leukaemia inhibitory factor.[85 86] Furthermore, there is a three-way cross-talk between stem cell and metabolic-immune genes including STAT-3 and PPARγ.[87] Other cytokines, such as IL-1, have been noted in opioid dependency[8] and are usually indicative of inflammasome activation, and fractalkine and MCP-1/CCL2 have also been found to be elevated and are relevant to neuropathological and atherogenic processes.[70] As TLR4 shares many ligands and cochaperones with the receptor for advanced glycation end products this auto-amplificatory system may also be involved.[88–91]

This study had various strengths and limitations. The application of sophisticated cardiovascular measurement technology to the problem of drug addiction is unique to this group. Extended follow-up and recruitment of ODP is also notoriously difficult. The statistical techniques employed here are advanced and robust. Design strengths include its complementary cross-sectional and longitudinal design. Limitations of the present work relate to difficulties quantitating frequently variable amounts of past drug use in the absence of a formal measurement instrument. Use of a formal instrument such as the addiction severity index, and spot tests for drugs and alcohol are indicated in future replications of this study. Employment of structural tests and mechanistic studies to investigate pathophysiological pathways, would be advisable in future iterations of this work.

It is important to remember that the models and results described in this work are not represented as reliable and generalisable calibrations of the cardiovascular effects of opioid exposure, in combination with other well-established risk factors. These models are the result of an empirical investigation of many correlated risk factors from a particular clinical cohort. It is most unlikely that similar model parameter estimates would be reproduced in a study of a different clinical cohort. The results serve a much more limited objective: is there evidence of an opioid effect on cardiovascular health, taking account of other known risk factors. We suggest that there is evidence for such an effect; although we must accept that this evidence is in part contingent on the adequacy of the model for these known risk factors. This is a general problem facing all studies of correlated risk factors, in the absence of a well-established mechanistic model. We would consider the impact of opioids to be established only if similar results were obtained in analysis of data from an independent case series. In this context the confirmatory nature of the present data to essentially identical results from the cross-sectional (only) data from Iran among men[16 17] and the various other findings noted in the literature review in the introductory section of the present report are particularly pertinent.

In summary, the present study documents elevated vascular stiffness and age associated with opioid use which is robust to multiple adjustments to other known risk factors. A strong dose–response effect was noted, and the duration of treatment administration was noted to be particularly important, being related to elevations in vascular age of the first, second, third and fourth order.

This finding indicates limitation of treatment duration wherever clinically possible. The effect was also worse in women. Overall this study raises many further mechanistic questions which indicate further study on this complex but important model condition.

**Acknowledgements** The authors would like to thank Dr Mervyn Thomas of Emphron for assistance with the statistics and graphical design.

**Contributors** The study was designed, the data collected, the analyses performed, the literature reviewed and the first draft written by ASR. GKH provided important intellectual input, assisted with study design, and revised and reviewed the manuscript. Both authors confirm the accuracy and integrity of the work and data collection and analysis.

**Funding** This research received no specific grant from any funding agency in the public, commercial or not-for-profit sectors.

**Competing interests** None.

**Ethics approval** Southcity Medical Centre Human Ethics Research Committee.

**Provenance and peer review** Not commissioned; externally peer reviewed.

**Data sharing statement** No additional data are available.

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
