## [Reviewer comments · BMJ Open]

Some articles will have been accepted based in part or entirely on reviews undertaken for other BMJ Group journals. These will be reproduced where possible.

ARTICLE DETAILS

TITLE (PROVISIONAL)	Impact of Lifetime Opioid Exposure on Arterial Stiffness and Vascular Age: Cross-sectional and Longitudinal Studies in Males and Females.
AUTHORS	Reece, Albert; Hulse, Gary

VERSION 1 - REVIEW

REVIEWER	Adrian Barnett Queensland University of Technology, Australia
REVIEW RETURNED	02-Jan-2014

GENERAL COMMENTS	A lot of work has gone into this paper, but more work needs to be done to make it readily understandable for the average BMJ Open reader. At the moment it seems like the statistical results are driving the paper, whereas it would be better if clear hypotheses drove the statistical models used. The title and motivation for the paper make an argument for the importance of gender, but the biological or social reasons behind gender are given only one line in the discussion. It feels as if the statistically significant results for gender are seen as the most important thing, rather than finding an explanation for the gender differences. The title could be simplified. Are these ideal controls for the cases? I recommend the paper "Compared to what? Finding controls for case-control studies" The Lancet, Vol. 365, No. 9468. (April 2005), pp. 1429-1433. To quote from that paper, "controls in a case-control study should represent those at risk of becoming a case". The cases come from the same clinic, which is good, but I would expect to see a discussion of their suitability and potential biases (this is also part of the checklist). The results would be better presented as differences between groups. Currently many of the results are presented using t-values or p-values. For example, on page 12, line 35, there is a factor estimate of 0.0719. Is this the difference between groups? It would be better to add the 95% confidence interval for the difference and remove the t- and p-values. This comment also applies to the tables. Some of the statistical models are very complex in terms of the independent variables used. For example, in table 5 there are five-way interactions between dose, duration, height, cigarettes and weight. It's hard to think of a clinical reason why these particular five variables would interact, as it is more likely that there are a statistical artifact from testing all possible interactions. In my experience it is
---

	very rare to fit anything beyond a three-way interaction, and any interaction needs to be carefully justified a priori. More interpretation needs to be given for the results. For example, page 14, 3rd paragraph, talks about power functions, but only talks about the statistical results and not what the non-linear associations look like or the clinical implications. Similarly on page 15 I'd prefer to see what the dose-response association looked like rather than the p-values. I really can't see the patterns described in Figure 7 (Page 13, 2nd paragraph). To me the lines look straight for women at around the age 30 mark. Also the start of the paragraph talks about a change point for women at 30 years, but then the end of the paragraph uses 55 years. Minor comments  - abstract and elsewhere, one or no decimal places would be adequate for the age statistics - abstract, the Radial Pulse Wave is not an intervention - abstract and elsewhere, multivariate means multiple dependent variables, you have used multiple variable regression - the first bullet point in the Article Focus section is contradictory - the last bullet point the the Key Messages section needs rewording, as the second part of the sentence does not follow from the first - The second bullet point states that the statistics methods are "Advanced", whereas I would say that all the methods used are now quite standard - page 6, line 56, why just quote the person years for 2009 only? - page 8, line 10, whilst this paragraph is well written, the language seems to be more suited to an opinion piece or editorial in my opinion - page 8, line 21, "means" not "implies" - page 8, line 38, "However a formal comparison of the effects by gender has not been possible in these earlier reports" That is surprising given that gender is almost always recorded - page 9, line 36, what defines baseline? Their first visit to the clinic? - page 9, line 46, what was excluded? the patients or their results for that visit? - page 11 " Random effects were assigned to unity and the patient identification code." This should come after the mention of the nlme model. Also, I don't understand what is meant by "assigned to unity"? Also, what was the motivation for using random effects? - page 11, was there a stopping rule for the least significant term? - page 12, line 9, these results are not in Table 1 - page 14, line 54, but the simpler model should be preferred on the principle of parsimony? - page 15, line 8, extra decimal place typo - as you have used some longitudinal data you should include a table of numbers at each visit and the reasons for drop-out. See the checklist for longitudinal studies. - Figure 1, is it right to call BMI and Ageing index? - Figure 3, the legend has the wrong labels - Figure 7, loess curves are used here but were not mentioned in the methods
--	---

REVIEWER	Takahiro Maeda
----------	----------------

	The Department of Community Medicine, Nagasaki University Graduate School of Biomedical Science, Japan
REVIEW RETURNED	07-Jan-2014

GENERAL COMMENTS	This case control study is cross-sectional and longitudinal study with 576 clinical controls and 687 opiate dependent patients over a total of 2,382 days. The author revealed that the body mass index (BMI) was lower in the opiate dependence (OD) group than that of control group. And in the longitudinal study, lower BMI seems stronger by the meaning of the effects of cumulative follow up days. The author made further analysis with BMI adjusted model and found that vascular reference age in participants with undergoing clinical management of their opiate dependence was higher than a function of chronologic ages and time. The significant change in vascular reference age and major arterial stiffness were observed only for women. Therefore, the author concludes that lifetime opiate exposure is an interactive cardiovascular risk factor particularly in females. This reviewer thought this article is basically treats interesting subject, but has serious limitations. First, because this was a case control study, more detailed characteristic of study population such as blood pressure, antihypertensive medications, BMI, nutrition status, drinking status, smoking status, serum HDL cholesterol, serum triglyceride, diabetes, estimate glomerular filtration rate, and menopausal status for women should be mentioned. Since social status was well-known factor that associated with various health problems (Shariful Islam SM, et al. BMC Public Health 2013; 13: 1217) and this study focused on the patients with opiate dependence, social status also might be important factor for this analysis. Authors should consider about the social status and add social status for this analysis. To prove that control group is suitable group as control sex-specific adjusted mean values or prevalence of classical cardiovascular risk factors for each group should be describe. Because the author found opiate dependence group was significantly lower BMI than that of control group and previous study reported "the stroke obesity paradox" (Andersen KK, et al. J Stroke Cerebrovasc Dis 2013; 22: 576-581) background of reduced BMI should be clarified. Author described that the significant association between lifetime opiate exposure and arterial stiffness was limited to female. Since menopausal status is well-known factor that has significant association with BMI stats and atherosclerosis, the data of menopausal status should be described.
--

	The author showed both of sex-combine values and sex-specific values. These presentations make little bit too heavy and make reader confused. Only sex-specific data is necessary because the author concluded sex-specific results. The author performed pulse wave analysis (PWA) on the right radial artery unless it was unavailable. However, PWA was well-known method with strongly associated with arteriosclerosis obliterans. Mean values of right and left side should be used. In statistic section, the author assigned as 88 when the individual vascular reference age returned as >80. However, this reviewer could not understand the evidence for this setting. In results section, the author declared that “racial stratification was not performed as our population is around 90% Caucasian in background”. The reviewer could understand that the stratification for this topic is not mandatory, but should be described as non-Caucasian in basal characteristics of study population and should compare the prevalence of opiate dependence group and control group.
--	---

REVIEWER	A Prof Mark Hutchinson University of Adelaide
REVIEW RETURNED	22-Jan-2014

GENERAL COMMENTS	the term opiate is not appropriate as this only refers to alkaloids contained in the opioid poppy. The term opioid captures the naturally occurring and more abundantly prescribed partially and fully synthetic opioid analgesics. Numbers of significant figures for % values are excessive and should be reduced to one sig figure. Numbers of sig figures for the remainder of the stats need to be reduced to 1 or 2 in most cases. Excessive use of abbreviations hampers the communication of the science. Where possible remove these (e.g. RA, ODP, OD etc) The information provided about opioid dependent populations and cardiovascular health is impressive, but can some comparison please be made regarding this group and opioid drug dependence and the rates of cardiovascular disease in another drug dependent population (e.g. alcohol, nicotine, methamphetamine or cocaine)? Additionally, what is the impact of lifestyle beyond opioid exposure that has contributed to this cardiovascular disease? The statement that we can draw parallels about ageing from the
---

	opioid drug dependent population is an exciting and provocative one. However, the concerns about lifestyle factors associated with drug abuse need to be addressed. The introduction provides significant evidence for the conduct of this study. However, this reviewer is left wondering how much of the cardiovascular damage is suggested to be mediated by exposure to the opioid, the many daily withdrawal, lifestyle, or an interaction of the above plus others? Can this be clarified? Please provide additional city, state and country details of the ethics committee to provide international context. Was any attempt made to include opioid of choice or dose in the statistical models? Figure 1 headings need to be converted from R column labels to interpretable values. While the tables 1-6 contain important information the majority of the readers will not know how to interpret these results, and the key findings are lost. Please consider revising down to just the key take home messages. Figures 1-8 also contain significant information, but as they are currently presented and described they hide many of the key take home messages. Importantly, the need to present all the transformations of the data are not clearly described. Much of the discussion is spent examining possible mechanisms to explain the interesting data. However, this goes beyond the important scope of this work. As such, greater emphasis needs to be provided to the interpretation of these data and positioning these data in the context of the studies mentioned in the introduction.
--	---

VERSION 1 – AUTHOR RESPONSE

Reviewer 1.

As suggested by this reviewer and by the editor, the title has been simplified.

Whilst this study compares two groups, one of cases and one of controls, it has not been designed as a “case control study” per se. I very much doubt that, with only 0.4% of the community addicted to heroin, a reasonable case could be made that certain non-opioid dependent patients were “at risk of becoming heroin dependent.” I am not aware that such criteria have been published to allow the selection of such a putative group. I have added a paragraph to the Discussion section which considers the degree to which the cases and controls are matched.

The presentation of the results in this paper is full and transparent, as is standard practice in innumerable other papers. It is understood that in presenting statistics, one presents the name of the test, and the key statistics from the test used, together with the applicable P-value. This is how data

is presented in Nature and Science and all leading journals. The factor estimate used on page 12 line 35 is the beta-factor used which emerges from the final multiple regression model which has been used. To clarify matters for this reviewer, I have added the S.E. of the estimate, which is also available in the final regression solution. The only reason I had not included it earlier was due the profusion of data in the results section. This is a problem which is alluded to subsequently by the second and third reviewers. However it would seem that a balance has to be reached between the detail with which the results are presented and the readability of those results, which may be termed the denseness or ease of understanding of the results which are presented. The manner of the present report represents a genuine attempt to strike a balance on this spectrum, according to conventions which have been tried and used many times previously.

The reviewer complains that some of our regression models are complex. This is an inevitable result of the question asked and the primarily statistical methodology employed to address it. This is a paper which asks a novel but, in view of the literature reviewed in the Introduction, an important question, namely, "*Is lifetime opioid exposure associated with measurable evidence of cardiovascular / organismal aging in both absolute and relative terms?*" The absolute question can be addressed relatively simply by the kinds of relatively straightforward analyses which are done in the earlier section of the Results. However the most important question at issue from a cardiological and epidemiological point of view is whether significant opioid exposure represents an additional cardiovascular risk factor in addition to known cardiovascular risk factors. This is a very different and mechanistically very important question. If opioid exposure is associated with stigmata of cardiovascular aging, but this can be accounted for simply by conventional risk factors, then it suggests that the elevated rate of cardiovascular disease seen amongst opioid dependent patients is related merely to increased exposure to known risk factors. However the demonstration that opioids account for variance in addiction to the known risk factors, suggests that more complex and possibly additional factors are at play which need to be accounted for. This has been done by what I would suggest has been a commendable effort to account for the known cardiovascular risk factors in our models. Hence the final multiple regression models which consider these questions exhaustively will inevitably be complex by nature of the question they are designed to investigate. I have now added an explicit statement to the relevant section of the Results that the particular terms chosen were indeed those which were significant in exploratory preliminary models.

Furthermore it would appear rather clear as to why there should be an interaction between many of the various terms. Height and weight will naturally interact and indeed the BMI is one formula which formalizes such interactions. The opioid dose-duration exposure is a standard way in which to account for lifetime opioid exposure, and has been used by WHO ¹ and NIH ² amongst others. Height is a key variable which relates to many techniques of measuring of arterial stiffness since it is related to the timing of arterial wave reflections, and it is hardly surprising that this remains in the final model ³⁻⁴. The reviewer's remark that "it is more likely that they are a statistical artefact from testing all possible interactions" is obviated by the remark that these were the most significant terms identified by the exploratory preliminary analyses. Again such techniques are used in countless studies of this type ⁵⁻⁶ and are in themselves unremarkable as indeed this reviewer subsequently correctly notes. However to clarify these matters I have added a remark to the relevant point in the results section that these issues are considered explicitly in the Discussion, and I have added remarks such as those above to the Discussion to this effect.

This reviewer wishes to see graphical representations of the models with power functions, in addition to the detailed statistical tests which have been presented. However there are already (now) six

figures in the main paper, and (now) three in the Supplementary data. Moreover a subsequent reviewer complains that there are in fact too many figures, and wants to see their number reduced in view of the density of the results presentation. In view of this conflict between the reviewers, we feel that the detailed statistical presentation provided adequately covers the issue. Moreover, since the variance in the various model AIC's is so tiny, 83.7 to 84.52, we do not feel that graphical display of these results would result in visually meaningful or discernible differences.

Comparison of the two sides of Figure 7 (males and females respectively) with loess curves fitted on the data shows clearly that the separation at the higher ages occurs to a much more marked extent amongst females, and it also begins much earlier. This graphical demonstration seems clear and unequivocal. This graphical separation is quantified by the statistical analysis presented in Results and Table 2.

This reviewer would appear to have misunderstood what is meant by the reference to age < 55 years. I have re-arranged this sentence now to make this clearer – when the analysis is restricted to women of ages <55 years the significant result indicated is obtained. This is also described in the title of Table 2. The need for this cut-off in the analysis may be related to the convex upward curvilinear nature of these data, and the higher cut-off age for analysis.

Minor Points

Ages – the significant figures have been reduced to one as suggested both in the Abstract and the Results.

Intervention. This has been corrected. The Intervention is now listed as Nil. The comment on methodology has been appended to the Design section.

Multivariate regression. This term has been changed to multiple regression throughout.

The reviewer's observation about this comment in the article Focus section is correct. However so too is the opening Focus remark. That is the motivation for this study. Despite a remarkable and striking concordance between several epidemiological studies on this area, any meaningful appreciation of this important observation is remarkably absent from the published literature. It is well known that stimulant abuse (tobacco, amphetamine and cocaine) are associated with arteriopathy and numerous cardiovascular complications, but this is not widely appreciated for opioid use. For this reason the subject bears further investigation using an endophenotype of what have mostly been epidemiological studies of cardiovascular-related mortality rates. The important question is "*Given the higher rate of mortality, is there an identifiable or quantifiable endophenotype that can be followed prospectively to study in detail what is presumably subclinical disease in these patients?*" That is the primary reason for this study. In this case the very important added bonus pertains however, since

cardiovascular age is the major surrogate marker for organismal age, which allows very important considerations of wider implications for organismal aging.

The third bullet point makes the point that power functions of opioid exposure duration enjoy improved fit to the model data, which is the major point made by the quoted UN document in relation to the link between tobacco exposure and cumulative lung cancer risk ¹. The second part of the sentence states that this finding underscores the importance of the exposure duration, which seems self-evident. In the context where opioid medication is frequently given for many decades, and indeed deliberately and determinedly so, this is a very important finding as it necessarily calls into serious question this common clinical standard and indeed major treatment goal, and suggests that the cumulative organismal dysfunction may be related to power functions of the total exposure duration. Therefore this is one of the major findings of this paper carries far-reaching implications for clinical practice.

The reviewer's comment about "advanced" statistical comments has been accepted. This word has been replaced with "detailed", and the sentence re-written slightly, as it is considered that detailed examination of this hypothesis has indeed been undertaken.

The reviewer has misunderstood this remark on page 6 line 56. Degenhardt's review was published in 2009 and the person-years relates to the whole of the period observed. I have therefore deleted the comment about the year of publication to remove any such possibility of misunderstanding.

The reviewer objects to the use of the word "exciting". This has now been removed. However I have to say that it is very exciting indeed. If opiate dependence is a model clinical laboratory for the observation of the accelerated accrual of organismal dysfunction and disease, which is what is consistently shown by the data across all organ systems, then this means that we have an important clinical laboratory in which to study the aging process in humans. This implies that a whole plethora of multi-system disease can be seen in such patients, and the inevitably improved understanding of the aging process can only benefit the remainder of the community. Whilst admittedly opioid dependency is a complex clinical disorder, when it is considered that most of the ageing research has been conducted in yeast, worms, flies and mice, the benefits of a robust human model for accelerated ageing seems very significant indeed. The reviewer and the editor are respectfully requested to ponder such implications for themselves. This conclusion seems both compelling and inescapable.

"Implies" has been changed to "means" page 8 line 21 in accordance with the reviewer's suggestion.

It was not possible to compare gender in the earlier single sex reports, as the data was limited to only one sex. This has now been made explicit by an addition to the end of this paragraph.

The reviewer is correct to understand that "baseline" relates to the initial assessment page 9 line 36. This is confirmed by the remark "Time in days was measured from the time of the first PWA study" on Page 10 line 10.

Page 9 line 46 I have clarified that the studies from that day were excluded, rather than those of the patient.

The order of the sentences at issue on Page 11 in Statistics has been reversed as recommended.

The motivation for using non-linear mixed effects modelling was, as clearly stated, time dependent repeated measures were performed on the same patients at multiple time intervals. Hence it is quite inappropriate to use simply multiple regression modelling. The specialist statistical advice to the project was that mixed effects modelling was the best way to do this.

Random effects were assigned to unity (1) and the patient case number, because that was the random effects of the model. The fixed effects were the dependent and independent variables as described. This was all done in accord with the specialized statistical advice to the project. As this reviewer correctly notes these statistical techniques are all very well accepted and not at all controversial.

No stopping rule was used for the statistical analysis. Model reduction halted when all terms or factors became significant.

The reviewer is correct that the age results are not in Table 1. The reference should be to Supplementary Table 1. This has now been corrected.

The reviewer's remark about parsimony is correct, but it adds nothing. Statistically the quadratic model is better than the linear, cubic or quartic model. From parsimony one would prefer the quadratic model to the combined linear-quadratic-cubic-quartic model – but in any event the quadratic model is superior. Therefore the main point that models with power functions in opiate duration are superior, is not perturbed.

The reviewer is thanked for picking up this typo in the decimal point. This has now been amended.

A new table has now been inserted, Supplementary Table 2 which provides the numbers of studies done in the various numbers categories.

Figure 1 - BMI varies with age, and so yes, it is indeed appropriate to call BMI an aging index.

The error in the legend label in Figure 3 has now been corrected. The reviewer is thanked for spotting this error.

The loess curves fitted to Figure 7 have now been mentioned in the Methods.

Reviewer 2.

The study compares two groups, control and opiate dependent. However it is not a case control study by design.

The reviewer's attention is drawn to Supplementary Tables 1 and 2 where most of the information he wishes to see is listed. That is blood pressure is shown in Supplementary table 2, Cholesterol and HDL and LDL and triglycerides, is shown in Supplementary Table 1, serum glucose is mentioned in Supplementary table 1, and serum creatinine is also mentioned in this table. The comparisons recommended by this reviewer have also been performed and are listed in Supplementary Tables 1 and 2.

The Methods section mentions that any patient with known cardiovascular disease is excluded. So no patients in this study were on antihypertensive medications.

The cause of the lower BMI in the opiate dependent patients is not known with any certainty. However the suggestion given on page 12 line 40-41 that this is due to the diversion of resources away from food and towards drug use seems reasonable, and fits well with what is known clinically about these patients.

Since we have no data on the hormonal status of these patients it is not possible to comment on any possible association with menopause. Opiate dependency is associated with hypothalamic hypogonadotrophic oligomenorrhoea and amenorrhoea. However I have added detailed data on the age distribution of this population assuming that 50 years is a mean age for menopause. Although it is not possible to make any definitive comment on this issue in the absence of data, it may be that the age-related hormonal suppression of the control group was balanced by the drug related hormonal suppression of the opiate dependent group. I have also added a comment to the Discussion section on this point which makes reference to the added data.

This reviewer complains that the presentation of the data and results analysis is too heavy and difficult to read. However the last reviewer complained that more data should have been given. It is

felt that the way the data has been presented, which accords with all the usual conventions for data presentation strikes a reasonable balance in this situation.

The cause of the absence of the right radial artery has now been specified. No cases of arteriosclerosis obliterans were encountered, and if they had have been they would have been excluded under the cardiovascular exclusions. As noted now, the causes of right radial artery absence which were encountered in this study were trauma, surgery or congenital absence.

Similarly no data was collected on racial background. Previous studies on this group of patients has found that 89% of our patients are of Caucasian background ⁷. For this reason other major reviews of Australian populations of drug dependent patients do not routinely include racial stratifications – see for example ⁸ and ⁹. However the reviewer quite correct that this data field should be added to future iterations of this work.

Reviewer 3.

The term “opiate” has now been changed to “opioid” throughout the manuscript.

I have reduced several percentages to one significant figure especially in the Abstract and Results, in accordance with the reviewer’s suggestion.

I have reduced the significant figures for statistics where this is appropriate. It is not very easy to do this with P-values for example, and S.E.’s of factor estimates (β -coefficients) need to be comparable to their variable β -coefficient estimates.

I have changed OD and ODP throughout the text to opioid dependence as recommended by this reviewer. The terms have however been retained in the abstract only, mainly because this helps to keep it within the required word count. It is however difficult to get away from RA in a manuscript of this nature.

Whist there is some information on other drug use by this cohort in the main databases, it is not complete enough, nor is it in a readily quantifiable form to allow indices of cardiovascular ageing to be calculated for other major drugs of abuse or addiction. However from the point of view of understanding drug-induced pathologies and age-related– and neuro- immunopathies particularly, the reviewer’s point is important, and would need to be addressed by further studies. We have preliminary data on age elevation in cannabis and amphetamine dependence, and these changes are

very marked indeed in these groups. However as the numbers in these groups are not very large, we have not considered to publish this work.

The issue with alcohol is more complex as it is a peripheral vasodilator, which will tend to reduce their apparent arterial stiffness and vascular age. Moreover as alcohol dependence becomes chronic and refractory cirrhosis often occurs. The increased levels of estrogens and cytokines associated with hepatic cirrhosis further causes peripheral vasodilatation and vascular shunting. Hence this technique is probably not well suited for the study of the acceleration of the aging processes in alcoholism. Other age related techniques such as telomere length in circulating PBL's, stem cell counts, cellular expression of senescence associated β -galactosidase or hippocampal shrinkage on volumetric MRI or hypothalamic inflammation (NF- κ B expression) and oedema may need to be followed to examine these effects in alcohol exposed individuals. The present datasets exclude anyone with either acute or chronic alcohol use. The data we have on this subject, which has not been presented, tends to confirm these findings consistent with peripheral vasodilatation despite what is obviously quite marked phenotypic age related changes in these patients.

The full cross-sectional database for this study has 158 data fields. As such it represents a very serious attempt to measure and quantify as many variables as possible. The point is that whilst a genuine attempt has been made to adjust for as many cardiovascular variables as practicable, the highly significant effect of opioid exposure, and dose-duration effect, and the power function for effect of exposure duration remain. To say this is due to some undefined "lifestyle factor" is to posit some variable for which has not been possible to control.

Interestingly this was the experience also of the Iranian authors, who performed similar multiple regression studies, and similarly found that when all traditional risk factors were accounted for there remained a significant effect which could only be attributed to opiate exposure^{5-6 10}.

The city, state and country details of the ethics committee have now been provided.

Yes the opioid dose has been included in many of the statistical models. This can readily be seen from inspection of the Results section and also Tables 3-6 and Supplementary Tables 8, 10, and 12.

No attempt was made to adjust for the primary opiate used. However the reviewer is referred also to reference¹¹ (reference 19 in the manuscript) where methadone was found to be the most severe opiate. A detailed analysis of dental disease made a similar finding against methadone¹².

Similarly an earlier study has detailed the effects of opiate withdrawal to elevate vascular age and arterial stiffness¹³ (reference 23 in the manuscript). This has now been made explicit in the final paragraph of the Introduction.

There are no right column labels in Figure 1.

The reviewer feels that Tables 1-6 are too complex and may complicate the presentation of the results. The reviewer is reminded that indeed this has already been done as far as possible. Whilst there are 6 tables presented in the body of the Report, there are in fact 13 tables presented in the Supplementary files. All pains have indeed been taken to trim the report to its bare bones and present only that data in the body of the manuscript which is most germane, and indeed essential to the main points of the paper.

Table 1 shows the exact statistical output of the comparisons of the age addiction effects by gender. Hence this is central to the major message of the paper (see Key Message 1).

Table 2 quantitates the very different graphical appearances of Figure 7, focussing on the age:addiction:sex interaction. Hence this is an important contribution to the major message of the paper and its analysis.

Table 3 presents important details on the power functions of opioid treatment duration, and quantitates the difference between the various models. Hence this is a major message of the paper (see Key Message 3).

Table 4 presents a detailed analysis of the all important dose-response relationships in the whole group and by sex. Hence it is difficult to see how this could be omitted.

Tables 5 and 6 present the full multiple regression models for the cross-sectional and longitudinal datasets respectively. This again is germane to the main points in the paper, and it is really not possible to combine, truncate or abbreviate these without detracting significantly from the rigor of the presentation of results.

This reviewer feels that the figures are also cumbersome, and hide important take home messages. The presentation of Results describes in considerable detail the difference in the BMI between the opioid dependent and control groups. Clearly this will complicate both the analysis and the presentation of results. It would be quite obvious to any physician or cardiologist that a different age related BMI trajectory necessitates careful consideration of this primary CVS risk factor for any age related analysis of a confounding chemical dependency. Several CVS parameters both for vascular age and for arterial stiffness are presented to demonstrate that the key parameters differ by opiate exposure in an age dependent manner.

The manuscript mentions that the C_AP_HR75 and C_AGPH_HR75 are the major measures of arterial stiffness, and analyses in some detail their age related differences, and in particular that the former rises in a linear fashion with age, while the latter is convex upwards and tends to plateau at

advanced ages. Since the Vascular age is a function of the C_AGPH_HR75 (also known as the Augmentation Index) this implies that it has a non-linear relationship with age, which must be borne in mind when considering studies of this type. All this is described in the paper.

The original manuscript also contains one Supplementary Figure. In deference to this reviewer, two of the figures have now been moved to Supplementary files.

Hence Figure 1 is quite logical and clearly sets out both the raw data, and the effects of age on the various adjusted parameters.

Figure 2 is very similar, it again adjusts for BMI for the above reasons, but includes time on the abscissa as it applies to the longitudinal dataset.

Old Figure 3 presents key age raw data by sex by age. This figure has now been moved to Supplementary Figure 1.

Old Figure 4 presents similar data to Figure 3 but the columns and key are reversed to allow direct comparison of each group by sex, which is one of the major foci of this paper. This figure has now been moved to Supplementary Figure 2.

Former Figure 5, now Figure 3, parallels Old Figure 3 but uniformly adjusts for BMI which shows better separation of the groups.

Former Figure 6, now Figure 4, repeats this exercise but again allows for detailed comparison by sex within each group.

New Figure 5 advances on the presentation of New Figure 4 by using loess curves for each group instead of regression lines of best fit, thereby making clear the different age: sex trajectory of the various parameters which is analyzed formally and quantitated in Table 2.

New Figure 6 presents the data by exposure groups which is standard in all such papers describing new risk factors for various diseases.

This meticulous form of data presentation has been employed to make clear the main messages of the paper and avoid criticism that only selected data are being presented, or that the data presentation is unclear or not transparent.

Hence the touchstones for the data presentation have been clarity, completeness, logical sequence, and unequivocal demonstration of age:sex:dependency differences, and the manner in which this has been used to guide the statistical analysis. I trust that the changes meet with this reviewer's approval.

Whilst it may be that this way of approaching results presentation and analysis is somewhat cumbersome, it is felt to be both clear and precise, which, given the clinical implications of this work, and the likely controversial nature of the present results, would appear to be a small price to pay for what is inherently a very complex subject.

So yes, it is felt that the sequence of results presentation is logical and sufficiently explained in the Results section, and that a nice flow from illustrative graphs to formal presentation of statistical results has been achieved with a meaningful interplay between text, graphs and tables.

In accordance with this reviewer's suggestion I have added a paragraph to the Discussion which demonstrates where the present report fits in the earlier reports which are cited in the Introduction.

It is not true that the Discussion focuses excessively on mechanistic processes. In its present iteration the Discussion section is just over five pages long (at 1.5 line spacing). Just over one page discusses mechanisms. I have added a caveat explicitly stating that this study is not designed as a mechanistic investigation. Given the significant body of literature on the cardiovascular complications of opioid dependence, it seems unlikely that this issue will be more widely recognized and considered and factored into clinical decision making (as noted by Reviewer 1) unless the several known mechanistic pathways begin to be understood and investigated, such that biological plausibility is widely seen to accompany the clinically increasingly characterized phenomenology.

However as noted in the Discussion section immune factors seem to be quite important. This was shown both in the First Supplementary Table, and also in the Final Supplementary Table 13 where the regression results of $\log(RA/CA)$ was shown. These points have now been remarked upon in a specific paragraph in the Discussion section.

References

1. Breslow N.E., Day N.E. *Statistical Methods in Cancer Research Volume II The Design and Analysis of Cohort Studies*. 1 ed. Lyon: International Agency for Cancer Research,, 1987.
2. Khademi H, Malekzadeh R, Pourshams A, Jafari E, Salahi R, Semnani S, et al. Opium use and mortality in Golestan Cohort Study: prospective cohort study of 50,000 adults in Iran. *BMJ (Clinical research ed)* 2012;344:e2502.
3. Nichols W.W., O'Rourke M.F., editors. *McDonald's Blood Flow in Arteries: Theoretical, Experimental and Clinical Principles*. London: Hodder Arnold, 2005.
4. O'Rourke MF, Pauca AL. Augmentation of the aortic and central arterial pressure waveform. *Blood Press Monit* 2004;9(4):179-85.
5. Sadeghian S, Dowlatshahi S, Karimi A, Tazik M. Epidemiology of opium use in 4398 patients admitted for coronary artery bypass graft in Tehran Heart Center. *J Cardiovasc Surg (Torino)* 2011;52(1):140-1.
6. Sadeghian S, Graili P, Salarifar M, Karimi AA, Darvish S, Abbasi SH. Opium consumption in men and diabetes mellitus in women are the most important risk factors of premature coronary artery disease in Iran. *International journal of cardiology* 2010;141(1):116-8.
7. Reece AS. Dentition of addiction in Queensland: poor dental status and major contributing drugs. *Australian dental journal* 2007;52(2):144-9.
8. Degenhardt L., Randall D., Hall W., Law M., Butler T., Burns L. Mortality among clients of a state-wide opioid pharmacotherapy program over 20 years: Risk factors and lives saved. *Drug Alcohol Depend* 2009;105(1-2):9-15.
9. Kelty E, Hulse G. Examination of mortality rates in a retrospective cohort of patients treated with oral or implant naltrexone for problematic opiate use. *Addiction* 2012;107(10):1817-24.
10. Sadeghian S, Darvish S, Davoodi G, Salarifar M, Mahmoodian M, Fallah N, et al. The association of opium with coronary artery disease. *Eur J Cardiovasc Prev Rehabil* 2007;14(5):715-7.
11. Reece A.S., Hulse G.K. Impact of Opioid Pharmacotherapy on Arterial stiffness and Vascular Ageing: Cross-sectional and Longitudinal Studies. *Cardiovascular Toxicology* 2013;13(3):254-66.
12. Reece A.S. Dental Health in Addiction. *Australian dental journal* 2009;54(2):185-86.
13. Reece AS, Hulse GK. Elevation of Central Arterial Stiffness and Vascular Ageing in Opiate Withdrawal: Cross-sectional and Longitudinal Studies. *Cardiovasc Toxicol* 2012;13(1):55-67.

VERSION 2 – REVIEW

REVIEWER	Adrian Barnett Queensland University of Technology, Australia
REVIEW RETURNED	26-Feb-2014

GENERAL COMMENTS	The authors respond to my previous concern about the controls in this case-control study by saying, this is not a case-control study 'per se', but it is clearly identified as a case-control study in the abstract. As with any case-control study, the potential biases of the controls used needs to be discussed. The authors have still not presented confidence intervals and cite Nature and Science in their defence. The BMJ Open guidelines are that "All articles should include the following [...] main results with (for quantitative studies) 95% confidence intervals and, where appropriate, the exact level of statistical significance". Confidence intervals (with associated means) are far more informative than t-values or p-values. The results include three- and four-way interactions between variables. Interactions of this size need to be justified biologically, otherwise they could easily be explained away by multiple testing. Alternatively such large interactions could be kept if they remain significant after cross-validation. I can understand how height would be important as a single variable in the final model, but some biological justification of how height can interact jointly with cigarettes, SP, dose and duration (taking just one example from Table 5). The response that "these were the most significant terms identified by the exploratory preliminary analyses" is no guarantee of them remaining significant in future samples and they could still simply be due to multiple testing. See, for example, the recent paper "Scientific method: Statistical errors" in Nature (vol 506, issue 7487). The authors use a similar AIC as a reason for not presenting the non-linear results on a meaningful scale. The AIC is a measure of overall model fit, which is a separate issue from the presentation of parameters of that model. "Random effects were assigned to unity and the patient identification code." I still don't understand what this means. I'm guessing it means that a random intercept was used for each patient.
--

REVIEWER	A/Prof Mark Hutchinson University of Adelaide, Australia
REVIEW RETURNED	03-Mar-2014

GENERAL COMMENTS	This revised manuscript addresses some of my concerns and is now much more accessible, but could still do with minor additional changes. For example the tables are still very long and labelling in the figures is not easily interpreted. Significant figures are still excessive. Whilst I commend the authors for their review of possible mechanisms that are contributing to the increased cardiovascular ageing I am still unsure if these need to be included here. This is an editorial decision.
---

	Minor points some of the new data expression have included too many sig figs examples of data presented with too many sig figs still persist in this manuscript Chi Squ needs to read Chi (with 2 super scripted) The figure references in the discussion appear to need updating. The term opiate has slipped back into the new sections of the discussion.
--	--

VERSION 2 – AUTHOR RESPONSE

Reviewer 1.

Whether this work can appropriately be considered as a case control study, really depends on the definition which is used of that term. As usually understood a case-control study sets out to match each case with a control, in various defined ratios. No suggestion occurs in the present manuscript that this methodology was attempted. The phrase “case control study” is not used in this manuscript. The phrase “case control” is used on one place only and that is describing a published study in the literature review. Both the abstract and the body of the report describe two parallel cohorts of cases and controls. The work might therefore properly be considered a parallel cohort study. In terms of the overall matching of groups, even case control studies usually set out to control mainly for age and sex. In the present study, although there was a four year difference in the ages of the two groups, this is not large, and in view of the age range studied, this is a small difference. The sex ratio of the two groups was not significantly different. Hence even if the study did purport to be a case control study (which it didn't) the matching of the two groups was quite reasonable.

It should also be noted that this is now the eighth paper in this series of cardiovascular papers. This body of work has been reviewed by 19 other referees (including the three involved in this paper). The suggestion that this is type of study is a case control study has not been made by any of the other 19 referees. Moreover I have published dozens of papers of this design, comparing many features, from IGF1, to immune, hepatic, metabolic factors, glycaemic indices, stem cells, cancer rates, dental illness, psychiatric disorders and greying of the hair in tow parallel cohorts as has been done in the present study. In none of these numerous cases have the numerous reviewers ever made a similar suggestion that what was undertaken was really a case control study. This implies that whilst the reviewer may be correct in some esoteric academic definition, his imputation of the meaning of this term is quite simply erroneous in the present context and this application.

Confidence intervals. The reviewer is thanked for pointing out this conflict with the journal standards. In accordance with this reviewer's suggestions, all the confidence intervals for the parameter estimates have been inserted in the tables in the appropriate place. Hence columns for lower and upper limits now largely replace previous columns for standard error of estimates and T-values. All the tables have been re-worked in this regard. They have therefore been completely deleted and replaced.

Multiple interactions. As noted above the major issue a report such as this needs to address is whether opiate exposure is significant after adjustment for all known cardiovascular risk factors, or if perhaps the putative effect of opiates might be mediated by other known risk factors which either co-vary with incidence, or co-vary by mechanism, with either the chemical effects of opioids themselves, or the accompanying lifestyle. Such procedures are quite standard in papers of this type. Models were typically trusted with the maximum number of independent variables they would tolerate, as guided by

preliminary testing of models, as stated earlier. This was done in several of the Iranian papers cited 1-3.

While this reviewer wishes to see more discussion of the pathophysiology, and the causal pathways by which the various final variables chosen might interact, I note that the following reviewer feels that there is already an excess of pathophysiological comment, which he wishes to see reduced. The requirements of the two reviewers would therefore appear to be in conflict in this respect. The independent variables chosen are all either recognized to be major determinants of cardiovascular health, or else relevant to this technique, as agreed by this reviewer in the case of height, which obviously impacts the timing and speed of wave reflection from the sites of peripheral reflection. Nor in fact is there obvious requirement from these considerations that the relationship of these various variables need necessarily be linear (and therefore additive) only. Since the relationship is likely complex, it might be expected that non-linear relationships such as potentiation, interactive effects, saturation effects, and this asymptotic effects may be displayed, and the analysis performed therefore makes not a priori assumptions about the nature of those interactions, as the present reviewer now implies.

The statistical adviser to this project has also made further advanced statistical points in relation to these considerations, and the reviewer and editor are cordially invited to consider these remarks.

Table 3 presents the results for the linear and non-linear models in profound detail for cross-sectional studies, and Supplementary table 11 does the same for the longitudinal models. There is therefore complete and full transparency of result reporting in this respect. The AIC's are described in Supplementary Tables 10 and 12, which, as noted by the reviewer, is the appropriate manner in which to compare model fits. Please note also the comments of the statistical adviser. If the reviewer is suggesting the further iterations of this work would find different interactions to be significant, I agree that that would almost inevitably occur. However the major results reported from our study are correct, and robust to various model fits in our dataset.

I have now added a remark in the Introduction that it was not our intention to establish a formal algorithm by which these various factors might be related.

“It was not our intention to establish a predictive algorithm by which these factors might be related.”
(Page 8).

Our intentions were much more conservative than that, as we set out only to show that the lifetime opiate exposure accounted for variance in the measured parameters beyond what was accounted for by inclusion of traditional cardiovascular risk factors. That is the reason for the preceding statement:

“The central investigative question to be prospectively assessed was whether there were changes in major central cardiovascular parameters in the opioid dependent group, and if so are they robust to adjustment for other confounding factors.”
(Page 8).

The reviewer's interpretation of the remarks in relation to the random intercept is indeed correct.

The editor will note the detailed and lengthy response from the statistical adviser to the project. In line

with his remarks two paragraphs have now been added, one to the end of the Results section and one penultimately immediately prior to the conclusion, and following on from the notes on study limitations.

The paragraph added to the end of the Results section reads:

“These overall results were broadly confirmed by the running of further analyses including by the use of by additivity and variance stabilization (AVAS) in R 4. In particular we analysed a range of models including different polynomial terms in known predictors, and matching models augmenting the known risk factors with heroin use. Models with and without heroin use were compared. These analyses confirmed that the model fit for the relationship between log (RA/CA) and the predictive variables was significantly improved by the inclusion of terms for opiate exposure (dose and duration) amongst the male opiate dependents, in the opioid dependent groups in both sexes and in all patients; that the relationship between the various model parameters was (unsurprisingly) unlikely to be of a strictly linear form; and that superior model fits for log (RA/CA) were likely to be accompanied by various higher order interactions between the most powerful predictive independent variables namely (log) SP, weight, height, tobacco consumption and opioid exposure dose and duration.”

The paragraph added following the limitations section reads:

“It is important to remember that the models and results described in this work are not represented as reliable and generalizable calibrations of the cardiovascular effects of opioid exposure, in combination with other well established risk factors. These models are the result of an empirical investigation of many correlated risk factors from a particular clinical cohort. It is most unlikely that similar model parameter estimates would be reproduced in a study of a different clinical cohort. The results serve a much more limited objective: is there evidence of an opioid effect cardiovascular health, taking account of other known risk factors. We suggest that there is evidence for such an effect; though we must accept that this evidence is in part contingent of the adequacy of the model for these known risk factors. This is a general problem facing all studies of correlated risk factors, in the absence of a well established mechanistic model. We would consider the impact of opioids to be established only if similar results were obtained in analysis of data from an independent case series. In this context the confirmatory nature of the present data to essentially identical results from the cross-sectional (only) data from Iran amongst males 2-3, and the various other findings noted in the literature review in the Introductory section of the present report are particularly pertinent.”

Reviewer 2.

This reviewer wants me to shorten the tables, but the first reviewer wants more detail presented. In this respect the desires of the two reviewers seem to be in irreconcilable conflict. In terms of table length, Tables 1 and 2 are quite brief with only 13 and 4 lines of data respectively, Tables 3, 4 and 5 are only half a page long. The sole exception is Table 6 which is probably giving rise to this perceived problem. However the results it presents are the most important in the paper in that they present the exact place of opioid exposure in the context of the other risk factors. Indeed in accordance with this reviewer's feeling the lengthy tables have therefore already been largely been consigned to the Supplementary material as far as this is reasonable and practicable.

This reviewer again expresses concern that there is about one page out of five of mechanistic

discussion. It is again reiterated that this was done also in Khademi's epidemiological report published in 2012 in the BMJ (344: e2502) 5. Furthermore it is noted also that some of this reviewer's papers also contain mechanistic discussion, even when they are reporting mainly phenomenological data.

To clarify matters I have added captions to the figures to make it clear what is being described. Please see the "Figure Captions" section.

The largest number of significant figures now appearing is four. All P values have been reduced to four significant figures in Tables and Results and Abstract.

Chi squ has been corrected to Chi2 throughout as requested.

The figure references in the Results and Discussion have been updated as requested. The Titles of all the figures have been changed in accordance with this Reviewer's directions.

All instances of "opiates" have again been corrected to "opioids" as requested.

References

1. Sadeghian S, Darvish S, Davoodi G, et al. The association of opium with coronary artery disease. *Eur J Cardiovasc Prev Rehabil*. Oct 2007;14(5):715-717.
2. Sadeghian S, Dowlatshahi S, Karimi A, Tazik M. Epidemiology of opium use in 4398 patients admitted for coronary artery bypass graft in Tehran Heart Center. *J Cardiovasc Surg (Torino)*. Feb 2011;52(1):140-141.
3. Sadeghian S, Grailli P, Salarifar M, Karimi AA, Darvish S, Abbasi SH. Opium consumption in men and diabetes mellitus in women are the most important risk factors of premature coronary artery disease in Iran. *International journal of cardiology*. May 14 2010;141(1):116-118.
4. Tibshirani R. Estimating transformations for regression via additivity and variance stabilization. *Journal of the American Statistical Association*. 1988 1988;83(402):394-405.
5. Khademi H, Malekzadeh R, Pourshams A, et al. Opium use and mortality in Golestan Cohort Study: prospective cohort study of 50,000 adults in Iran. *BMJ (Clinical research ed)*. 2012;344:e2502.